# Decoding WW domain tandem-mediated target recognitions in tissue growth and cell polarity

Zhijie Lin[1†], Zhou Yang[1†], Ruiling Xie[2,3], Zeyang Ji[1], Kunliang Guan[2], Mingjie Zhang[1,4*]

[1]Division of Life Science, State Key Laboratory of Molecular Neuroscience, Hong Kong University of Science and Technology, Hong Kong, China; [2]Department of Pharmacology and Moores Cancer Center, University of California, San Diego, La Jolla, United States; [3]Department of Otolaryngology, Head and Neck Surgery, Peking University First Hospital, Beijing, China; [4]Center of Systems Biology and Human Health, Hong Kong University of Science and Technology, Kowloon, China

**Abstract** WW domain tandem-containing proteins such as KIBRA, YAP, and MAGI play critical roles in cell growth and polarity via binding to and positioning target proteins in specific subcellular regions. An immense disparity exists between promiscuity of WW domain-mediated target bindings and specific roles of WW domain proteins in cell growth regulation. Here, we discovered that WW domain tandems of KIBRA and MAGI, but not YAP, bind to specific target proteins with extremely high affinity and exquisite sequence specificity. Via systematic structural biology and biochemistry approaches, we decoded the target binding rules of WW domain tandems from cell growth regulatory proteins and uncovered a list of previously unknown WW tandem binding proteins including β-Dystroglycan, JCAD, and PTPN21. The WW tandem-mediated target recognition mechanisms elucidated here can guide functional studies of WW domain proteins in cell growth and polarity as well as in other cellular processes including neuronal synaptic signaling.
DOI: https://doi.org/10.7554/eLife.49439.001

*For correspondence:
mzhang@ust.hk

†These authors contributed equally to this work

## Introduction

WW domain, one of the smallest protein-protein interaction domains composed of only ~35 amino acid residues, are widely distributed in many proteins with diverse functions (e.g. a totally of 95 WW domain in 53 proteins in the human proteome). WW domains organize molecular assemblies by binding to short, proline rich peptide motifs each with ~4–6 residues (*Sudol and Hunter, 2000*; *Salah, 2012*). The type I WW domains bind to a consensus 'PPxY' motif (where P is proline, x is any amino acid and Y is tyrosine, also known as PY-motif) with very modest affinities (with $K_d$ in the range of a few to a few tens of μM) (*Chong et al., 2010*; *Kato et al., 2002*; *Kato et al., 2004*; *Aragón et al., 2011*). The human proteome contains ~1,500 'PPxY' motifs in >1000 proteins (*Hu et al., 2004*; *Tapia et al., 2010*). Most of the WW domains in the human proteome belong to the type I (52 out of a total of 95). Therefore, the combinations of potential WW domain/'PPxY' interactions are enormous. This immediately raises an issue on WW domain-mediated target binding specificities.

Taking the Hippo signaling pathway for an example, the pathway is organized by serval WW domain proteins (e.g. YAP, TAZ, KIBRA, and SAV1) and a number of PY-motif containing proteins such as LATS (LATS1 and LATS2), Angiomotins (AMOTs, including AMOT, AMOTL1 and AMOTL2), and PTPN14 (*Pan, 2010*; *Sudol, 2010*; *Salah and Aqeilan, 2011*; *Yu and Guan, 2013*) (*Figure 1A*). The interactions between WW domains and PY-motifs in the Hippo pathway are very promiscuous,

**Figure 1.** Super strong bindings of the KIBRA WW tandem to PY-motif containing proteins in cell growth control. (**A**) Schematic diagram showing the domain organization of selected WW-tandem containing proteins and multiple PY-motifs containing proteins in cell growth and polarity. (**B**) ITC-derived binding affinities of WW tandems from KIBRA and YAP in binding to PY motifs from PTPN14, Dendrin, AMOT and LATS1. (**C–D**) Example ITC titration curves showing the bindings of KIBRA WW12 to PTPN14 PY12 (**C**) and to Dendrin PY23 (**D**). Due to the very tight direct bindings between the

*Figure 1 continued on next page*

*Figure 1 continued*

WW tandem and each of the PY-motifs, we used a competition-based binding assay (i.e. by first saturating each PY-motifs with YAP WW12 which has a modest binding to the PY-motif) to obtain reliable dissociation constants. (**E–G**) ITC-based measurements of the bindings between WW tandems of KIBRA and YAP and PY motifs of AMOT and Expanded. Both PY motifs of AMOT and Expanded are separated by long and flexible linkers. For simplicity, we have only listed $K_d$ values (showed as values ± fitting errors) for the ITC experiments in the figure. Other thermodynamic parameters (i.e. $K_d$, $\Delta H$, $T\Delta S$, and N value) of each ITC titration experiment covered in this study are compiled and presented in *Supplementary file 2*. The exact sequence of the WW tandems and its binding targets investigated in this study is compiled in *Supplementary file 4*.
DOI: https://doi.org/10.7554/eLife.49439.002

The following figure supplements are available for figure 1:

**Figure supplement 1.** Super strong bindings of the KIBRA WW tandem to PY-motif containing proteins functioning in cell growth control.
DOI: https://doi.org/10.7554/eLife.49439.003

**Figure supplement 2.** ITC experiments showing the bindings of Dendrin PY23 or β-DG PY34 to KIBRA WW12, showing that the TRX-tag had minimal impact on the bindings of each of the two peptides to the KIBRA WW12 tandem.
DOI: https://doi.org/10.7554/eLife.49439.004

as nearly everyone of these WW containing proteins has been reported to interact with anyone of the PY-motif containing targets. For example, YAP WW domains have been reported to bind to PY motifs from LATS, AMOTs, PTPN14, p73, SMAD1, etc. (*Hao et al., 2008*; *Oka et al., 2008*; *Zhang et al., 2008*; *Chan et al., 2011*; *Wang et al., 2011*; *Zhao et al., 2011*; *Wang et al., 2012*; *Huang et al., 2013*; *Liu et al., 2013*; *Strano et al., 2001*; *Alarcón et al., 2009*; *Yi et al., 2013*; *Michaloglou et al., 2013*). KIBRA WW domains have also been reported to bind to PY-motifs from LATS, AMOTs, and PTPN14 (*Baumgartner et al., 2010*; *Genevet et al., 2010*; *Yu et al., 2010*; *Knight et al., 2018*; *Xiao et al., 2011*; *Hermann et al., 2018*; *Wang et al., 2014*). The WW domains of SAV1 can bind to PY motifs of LATS (*Tapon et al., 2002*). Additionally, PY motifs of LATS1, AMOT and PTPN14 have also been shown to bind to WW domain-containing NEDD family E3 ligases (*Kim and Jho, 2018*; *Nguyen and Kugler, 2018*; *Salah, 2012*). However, it is not clear how such a large array of WW/PY-motif interactions in the regulation of Hippo signaling are inter-related during cell growth processes and whether all these reported interactions occur in living cells. Since whether YAP is in nuclei or in cytoplasm dictates the fate of cell growth and polarity (*Sun and Irvine, 2016*; *Moya and Halder, 2019*; *Fulford et al., 2018*; *Yu et al., 2015a*), it is envisaged that at least some of the WW/PY-motif interactions need to be specific.

In cell growth and polarity regulations, YAP sits at the final converging point integrating both Hippo pathway-dependent and -independent upstream signals and regulating gene transcriptions via its WW tandem-mediated binding to PY-motif containing proteins including LATS, AMOTs and PTPN14 (*Sun and Irvine, 2016*; *Moya and Halder, 2019*; *Fulford et al., 2018*; *Yu et al., 2015a*). KIBRA also contains a WW tandem that binds to PY-motifs from the same set of proteins. Additionally, the cell polarity regulator, MAGIs (MAGI1-3; *Figure 1A*), each contains a WW domain tandem capable of binding to various PY-motif proteins such as LATS, AMOTs and PTPN14 (*Couzens et al., 2013*; *Wang et al., 2014*; *Bratt et al., 2005*; *Patrie, 2005*). A systematic and quantitative study of the interactions between the three WW tandem proteins (YAP, KIBRA, and MAGIs) and many of the PY-motif containing proteins in cell growth control (e.g. those highlighted in *Figure 1A*) will be vitally important to understand how these proteins function to orchestrate cell growth control.

In this study, we decoded the target binding mechanisms governing the WW tandem-mediated target interactions by focusing on proteins involved in cell growth and polarity regulations. To achieve this goal, we determined the high resolustion structures of 7 representative WW tandem/target complexes, and measured the binding constants of >100 WW domain tandems or isolated WW domains binding to known or previously unknown targets. In sharp contrast to the common perception, we discovered that the WW tandem and PY-motif interactions are often extremely specific and strong, a finding that can rationalize numerous protein-protein interactions in cell growth and polarity. The rules underlying WW tandem/PY-motif interactions revealed in our study also allowed us to predict previously unknown binders of WW domain proteins. The results presented in this study may serve as a portal for future studies of WW domain containing proteins in general.

## Results

### WW Tandem-mediated bindings to PY-motif proteins can be extremely specific and strong

To begin such a systematic study, we first set out to compare bindings of the WW tandems of YAP and KIBRA with PY-motifs from LATS, AMOTs and PTPN14. We also included another PY-motif containing protein Dendrin in the study, serving as a reference for super strong target bindings involving WW domains ($K_d$ ~2.1 nM, *Figure 1B and D*; also see our recent work by *Ji et al., 2019*). Similar to the two PY-motifs of Dendrin that are separated by only two residues, the first two PY-motifs of PTPN14 are also next to each other with only two residues in between, and the PY12 motif sequence is evolutionarily conserved (*Figure 1—figure supplement 1A*). In contrast, PY3 and PY4 of PTPN14 are separated by 178 residues. We found that the KIBRA WW tandem also binds to PTPN14 PY12 with a super strong affinity ($K_d$ ~8.2 nM) (*Figure 1B and C*). We were not able to perform quantitative binding assay between KIBRA WW tandem and PTPN14 PY34 as we could not obtain purified recombinant PY34. In a pull-down-based assay, the interaction between the KIBRA WW tandem with PY34 was essentially undetectable when using PY12 as the control (data not shown). We further showed that KIBRA WW1 had a very weak binding to PTPN14 PY12 ($K_d$ ~32 μM) and KIBRA WW2 had no detectable binding to PTPN14 PY12 (*Figure 1—figure supplement 1B and C*). Taken together, the above biochemical study revealed that the KIBRA WW tandem, via synergistic binding to the two PY-motifs immediately next to each other (i.e. PY12), can form a very tight complex with PTPN14.

The very tight bindings of the two closely spaced PY-motifs of PTPN14 (PY12) and Dendrin (PY23) with the KIBRA WW12 tandem (*Figure 1A and B*) prompted us to analyze the PY-motif sequences of AMOTs and LATS carefully. We found that the last canonical PY-motif of AMOT (the 'PPEY'-motif, named as PY3 in the literature) is preceded with an 'LMRY'-motif, and these two motifs are only separated with two residues (*Figure 1A*, and we define these two motifs as 'PY34' from hereon). This sequence is conserved for AMOT and AMOTL1 throughout the evolution (*Figure 1—figure supplement 1A*). We also found that the second canonical PY-motif of LATS1 ('PPPY') is preceded with an 'APSY'-motif, and the two motifs are separated by only three residues with the sequence of '$^{552}$QGP$^{554}$' (*Figure 1A*, and these two motifs are defined as 'PY23'). Again, the sequence containing the 'APSY'- and 'PPPY'-motifs and the gap residues in between are highly conserved in LATS1 in different species (*Figure 1—figure supplement 1A*). Interestingly, a Gly553Glu mutation of LATS1 was found in patients with renal cell carcinoma (*Yu et al., 2015b*). Unexpectedly, the KIBRA WW tandem binds to AMOT PY34 with a high affinity ($K_d$ ~96 nM), although the PY3 sequence severely deviates from the canonical WW domain binding PY-motif of 'PPxY'. It is important to note that the KIBRA WW tandem binds to AMOT PY12, of which both PY1 and PY2 are canonical PY-motifs but are separated by 130 residues, with a very weak affinity ($K_d$ ~16.7 μM) (*Figure 1E*). We further found that the KIBRA WW tandem binds to LATS1 PY23 with a $K_d$ of ~0.78 μM (*Figure 1B*), but much more weakly to the canonical PY1 motif ($K_d$ ~78 μM; *Figure 1—figure supplement 1F*). The above biochemical analysis revealed that the two closely spaced PY-motifs in both AMOT and LATS1 can bind to the KIBRA WW tandem with high affinities, although the first PY-motif of both proteins deviates from the canonical PY-motif sequence.

The two WW domains of YAP are also arranged next to each other forming a WW domain tandem (*Figure 1A*). Like the isolated WW1 in KIBRA, isolated WW1 or WW2 of YAP can bind to PY-motif sequences with $K_d$ values of a few μM or a few tens of μM (*Figure 1—figure supplement 1B and D*). Very surprisingly, the YAP WW tandem binds to closely spaced PY-motifs of PTPN14, AMOT, and Dendrin with affinities of a few hundred to a few thousand fold weaker than the KIBRA WW tandem does (*Figure 1B*). These results revealed that WW domain tandem-mediated protein-protein interactions among proteins shown in *Figure 1A* are extremely specific, a finding that is totally unexpected and will have immense implications in understanding the protein interactions and their functions in cell growth and cell polarity.

### Structures of the KIBRA WW tandem in complex with PY-motifs

We next pursued detailed structural studies trying to understand the mechanistic bases governing the vast affinity differences between the WW tandems of KIBRA and YAP in their bindings to the

common set of PY-motif proteins shown in *Figure 1A*. We solved the high-resolution crystal structures of KIBRA WW tandems in complex with PTPN14 PY12 (*Figure 2A*), AMOT PY34 (*Figure 2B*) and LATS1 PY23 (*Figure 2C*, *Figure 2—figure supplement 1* and *Supplementary file 1*).

The structures of the WW tandems in the three complexes are extremely similar, with the overall pairwise RMSD values in the range of 0.63–1.0 Å (*Figure 2D* and *Figure 2—figure supplement 1A*). Every WW domain forms a canonical β-sheet fold with three anti-parallel β-strands. The two WW

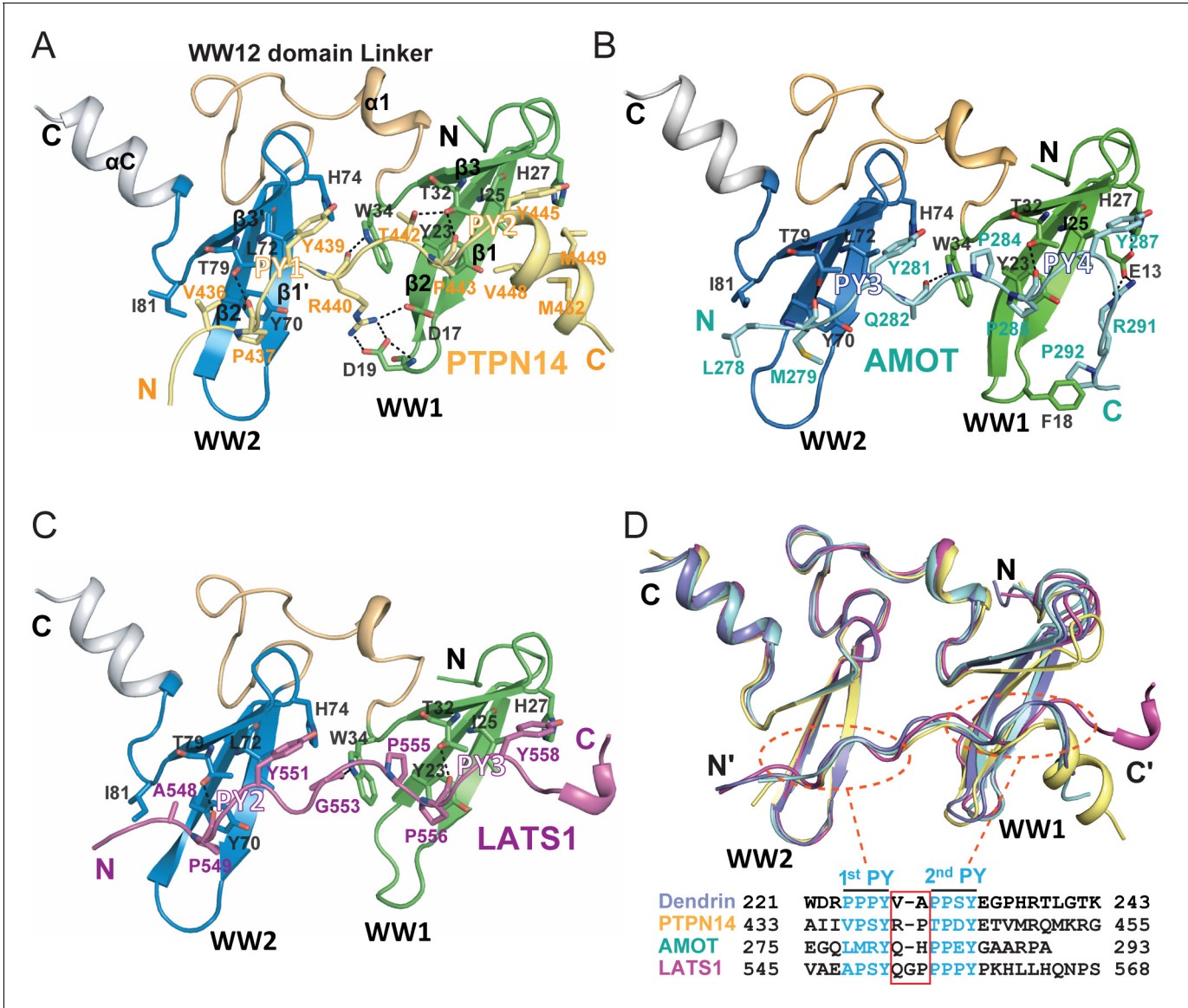

**Figure 2.** Structures of the KIBRA WW tandem in Complex with various closely spaced PY motifs. (**A–C**) Combined ribbon and stick model showing the structures and detailed interactions of the KIBRA WW tandem in complex with PY motifs from PTPN14 (**A**), AMOT (**B**), and LATS1 (**C**). In the crystals of the KIBRA/AMOT complex, WW12 adopts a domain-swapped dimer similar to the crystals of the KIBRA/Dendrin complex described earlier (*Ji et al., 2019*), so the KIBRA/AMOT supramolecular complex is formed from two different polypeptides in this structure. However, in solution KIBRA and AMOT form a 1:1 complex. (**D**) Superimposition of the structures of the KIBRA WW tandem in complex with PY motifs from PTPN14, AMOT, LATS1, and Dendrin. The amino acid sequences of the PY motifs are numbered and aligned at the bottom of the panel, and the red box highlights the 2–3 residue linking sequences of the PY motifs. The color-coding scheme for the structures is used throughout the manuscript.

DOI: https://doi.org/10.7554/eLife.49439.005

The following figure supplement is available for figure 2:

**Figure supplement 1.** Structures of KIBRA WW12 in Complex with various closely connected PY motifs.

DOI: https://doi.org/10.7554/eLife.49439.006

domains align with each other in a head-to-tail manner, so that the two PY-motif binding pockets of the WW tandem are positioned right next to each other to accommodate the two PY-motifs in each of the three targets. The head-to-tail orientation of the WW tandem imposes a restriction that the target peptide can only bind to the WW tandem in an antiparallel manner (i.e. 1st PY binds to WW2 and 2nd PY binds to WW1; *Figure 2D*). Though the direct contacts between the two WW domains in the tandem are minimal, the WW tandem forms a stable structural supramodule by extensive interactions between the inter-domain linker (residues I35-L57) and an α-helix extension (αC) immediately following WW2 via hydrophobic, charge-charge and hydrogen bond interactions (detailed in *Figure 3A*). Perturbations of the supramodule formation (e.g. single point mutations such as I35D, F47A, L57D, and W88A, or deletion of αC) invariably weakened the binding of the WW tandem to its targets (*Figure 3B*), indicating that formation of the WW12 supramodule is critical for high affinity bindings to the three targets studied here as well as to the PY23 of Dendrin (*Ji et al., 2019*).

The binding between WW1 and the second PY-motif of the three targets, each has a classical 'ΨPxY'-motif (where 'Ψ is an aliphatic or small polar amino acid with Pro being preferred, *Figure 2D*), almost perfectly fits into the optimal type I WW/target binding (*Figure 2A–C*) (*Macias et al., 1996*). The interactions between WW2 and the first PY-motif are more variable, as their sequences can significantly deviate from the optimal 'PPxY'-motif (e.g. AMOT's 1st PY motif has an unusual 'LMRY' sequence, *Figure 2D*). Correspondingly, KIBRA WW2 has an Ile (I81) instead of a highly conserved Trp at the end of the β3-strand (e.g. W34 in WW1; *Figure 2A–C*). The Leu and Met in the PY3-motif of AMOT form hydrophobic interaction with I81 and Y70 from WW2. This explains why KIBRA WW2 can accommodate non-'ΨPxY' motif sequences, as long as the two residues corresponding to 'Ψ' and 'P' are hydrophobic. The fact that the type I WW domain can bind to a 'ΦΦxY' sequence motif (where 'Φ' can be other uncharged amino acids in addition to Pro; for example 'VPSY' and 'TPDY' in PTPN14, 'APSY' in LATS1, and 'LMRY' in AMOT) significantly expands the target binding repertoire for WW domain-containing proteins.

In addition to the two PY-motifs, the residues in the linker and the C-terminal extension of the targets also contribute to their bindings to the KIBRA WW tandem. A residue in the PY-motif linker (R440 in PTPN14, Q282 in AMOT and G553 in LATS) uses its carbonyl group to form a hydrogen-bond with the sidechain of Trp34 (*Figure 2A–C*). The sidechain of R440 of PTPN14 also forms charge-charge interactions with D17 and D19 in β12 loop of WW1 (*Figure 2A*). The C-terminal extension of PTPN14 and that of AMOT form additional hydrophobic and charge-charge interactions with the WW1 domain (*Figure 2A and B*). All these interactions contribute to the bindings of PTPN14 and AMOT to KIBRA WW tandem.

## The WW tandems of MAGI2, MAGI3 and KIBRA resemble with each other in target bindings

Many WW domain proteins contain multiple WW domains connected in tandem (*Figure 3—figure supplement 1*). We asked whether there might exist type I WW tandems adopting similar high affinity target bindings as KIBRA WW tandem does. Since the inter-domain linker and the α-helix following WW2 are critical for forming the supramodular structure and target bindings of KIBRA WW tandem, we searched for possible existences of these two elements in other type I WW tandems (*Figure 3C*). By sequence alignment of WW tandems, including those from KIBRA, MAGIs, YAP, SAV1, WWOX, ITCH WW12 and ITCH WW34, we found that MAGI2 and MAGI3 WW tandems have striking similarities to KIBRA WW tandem both in the inter-domain linker (23 aa in MAGI2 and MAGI3 vs 24 aa in KIBRA and with ~60% sequence similarity) and in the C-terminal helix regions (*Figure 3C* and *Figure 3—figure supplement 2A*). To our delight, ITC assays showed that MAGI2 and MAGI3 WW tandems indeed bind to PTPN14 PY12 with strong affinities ($K_d$ values of ~55 nM and ~103 nM, respectively, *Figure 3D*). It is noted that although MAGI1 WW tandem shares a similar C-terminal helix extension with the KIBRA WW tandem, it possesses a longer WW12 domain linker (*Figure 3C*). Correspondingly, MAGI1 WW tandem was found to bind to PTPN14 PY12 with a modest affinity ($K_d$ ~4 μM; *Figure 3D*), presumably due to the spatial hindrance caused by the extra amino acids in the linker. We further found that MAGI2 and MAGI3 WW tandems bind to Dendrin PY23 with extremely high affinities ($K_d$ values of ~2.3 and~3.6 nM, *Figure 3—figure supplement 2B*), further supporting the conclusion that MAGI2, MAGI3 and KIBRA WW tandems share a high similarity in target bindings. Like KIBRA WW12, MAGI2 and MAGI3 WW tandems bind to LATS1 PY23 with modest affinities (*Figure 3D*). Finally, the rest of the type I WW tandems analyzed in

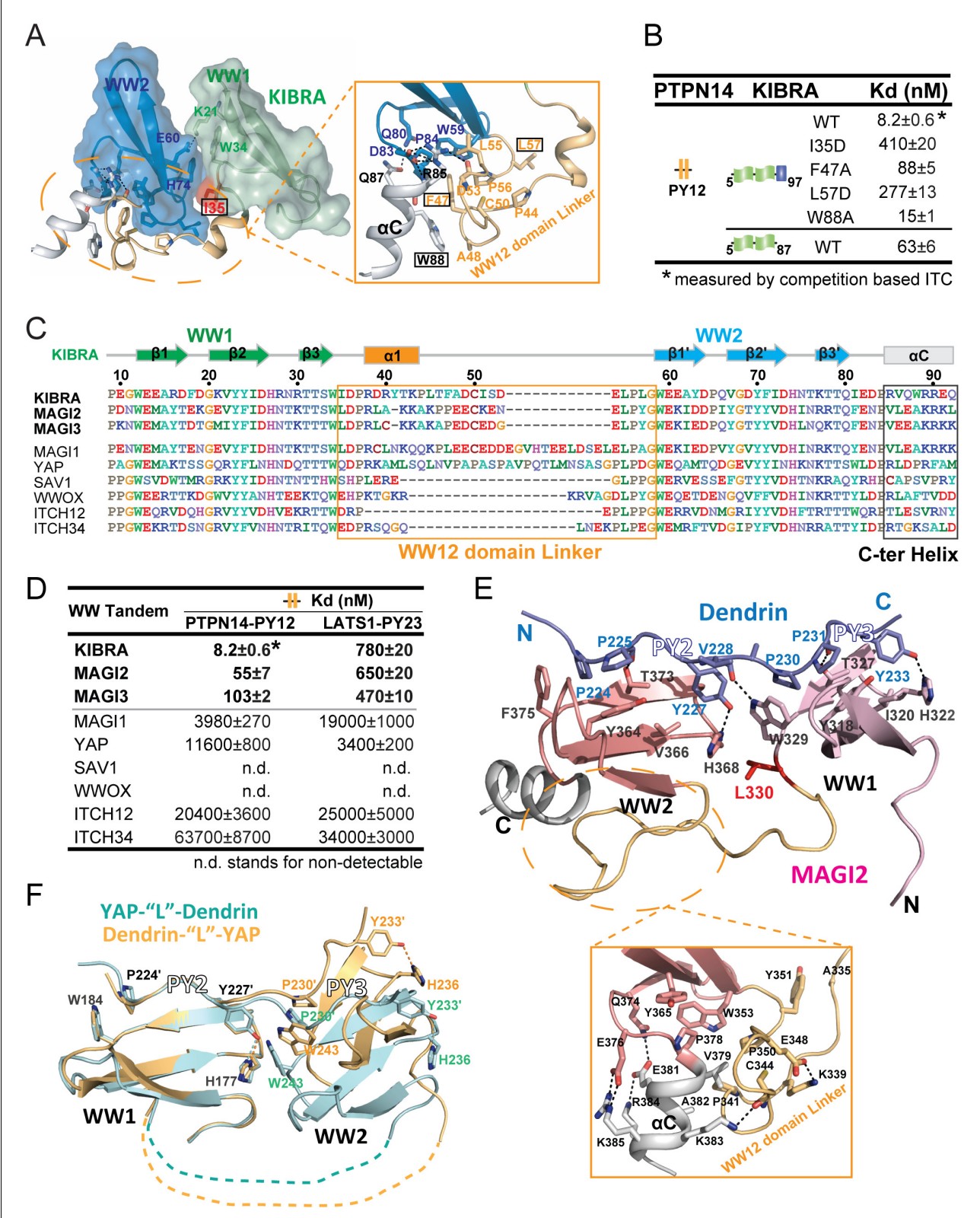

**Figure 3.** Structural basis governing the strong bindings of various WW tandems to their PY-motif containing targets. (**A**) Combined surface and ribbon-stick model showing the structural basis for the formation of the KIBRA WW tandem supramodule. The detailed interactions between the inter-domain linker and C-terminal α-helix (αC) are shown in an expanded insert at right. The residues chosen for mutational analysis in Panel B are boxed. (**B**) ITC-based measurements comparing the binding affinities of the KIBRA WW tandem and its various mutants to PTPN14 PY12. (**C**) Amino acid

*Figure 3 continued on next page*

*Figure 3 continued*

sequence alignment of the WW tandems from KIBRA, MAGI1, MAGI2, MAGI3, YAP, SAV1, WWOX and ITCH-WW12/WW34, showing that the WW tandems of KIBRA, MAGI2, and MGAI3 are more similar to each other. (D) ITC-derived affinities of various WW tandems binding to PTPN14 PY12 and LATS1 PY23. (E) Combined ribbon and stick diagram showing the structure and detailed interaction of the MAGI2 WW tandem in complex with Dendrin PY23. The detailed interactions between the inter-domain linker and C-terminal α-helix are shown in an expanded box below the structure. (F) Ribbon and stick model showing the superimposition of two versions of the structures of YAP WW tandem in complex with Dendrin PY23. The structure colored in brown was obtained by fusing Dendrin-PY23 to the N-terminus of the YAP WW tandem, and the one colored in cyan was obtained by fusing Dendrin-PY23 to the C-terminus of the YAP WW tandem. The structures are superimposed by overlaying the WW1, showing different orientations of WW2 between the two structures. The dotted lines denote the linking sequence between WW1 and WW2, which could not be traced in the two crystal structures.

DOI: https://doi.org/10.7554/eLife.49439.007

The following figure supplements are available for figure 3:

**Figure supplement 1.** Domain organization of Type I WW containing human proteins.

DOI: https://doi.org/10.7554/eLife.49439.008

**Figure supplement 2.** Structural basis governing the strong bindings of various WW tandems to their PY-motif containing targets.

DOI: https://doi.org/10.7554/eLife.49439.009

*Figure 3C* were found to bind to PTPN14 PY12 or LATS1 PY23 with very weak affinities or with no detectable binding (*Figure 3D*). Taken together, the above biochemical and bioinformatics analyses strongly indicated that WW tandems can have very strong affinities and exquisite specificities in recognizing their targets.

We solved the crystal structure of MAGI2 WW tandem in complex with Dendrin PY23 at the 1.65 Å resolution (*Figure 3E* and *Supplementary file 1*). A structural comparison of the MAGI2 WW tandem in complex with Dendrin with those of the KIBRA WW tandem in complex with various ligands shown in *Figure 2* showed that the overall structures of KIBRA and MAGI2 WW tandems in these complexes are similar to each other (RMSD values of 2.3–2.7 Å for the Cα atoms, *Figure 3—figure supplement 2C*). Notably, the extensive interactions between the inter-domain linker and the C-terminal helix stabilize the supramodular structure of the MAGI2 WW tandem, and the Dendrin PY23 binds to the MAGI2 WW tandem following almost the exactly same detailed interactions as those observed in the KIBRA WW12/Dendrin PY23 complex (*Figure 3E*) (*Ji et al., 2019*).

## YAP WW tandem adopts a very different structure and target binding mode compared to the KIBRA WW tandem

We also tried to determine the structures of the YAP WW tandem in complex with various ligands listed in *Figure 1B* to understand why YAP binds to these ligands with much lower affinities than does the KIBRA WW tandem. Despite extensive trials, we were not able to obtain any crystals when complexes were prepared by mixing the YAP WW tandem with the ligand peptides, likely due to conformational flexibilities of the WW tandem (see below). We were able to obtain high diffraction quality crystals when the Dendrin PY23 sequence was fused to either the N- or C-termini of the YAP WW tandem. We solved the structures of these two versions of the complex (*Figure 3F*, *Figure 3—figure supplement 2D* and *Supplementary file 1*). Although each WW domain engages the 'PPxY' motif following the canonical type I WW/target bindings (*Figure 3—figure supplement 2D*), the overall structure of the YAP WW tandem/Dendrin PY23 complex differs substantially from those of the KIBRA/MAGI2 WW tandems in complex with the same ligand (*Figure 3E* vs *Figure 3F* for example) in two distinct aspects. First, the inter-domain linker of the YAP WW tandem is unstructured and there is no C-terminal helix following YAP WW2. Thus, the two WW domains of YAP do not form a supramodule due to very limited inter-domain contact. This is supported by the two versions of the complex structures solved, in which the inter-domain orientations are obviously different (*Figure 3F*). Second, the Dendrin PY23 binds to the YAP WW tandem in a parallel orientation (i.e. PY2 binds to WW1 and PY3 binds to WW2), instead of the antiparallel binding orientation observed for the KIBRA/MAGI2 WW tandems. This is again likely attributed by the lack of inter-domain interactions between the two WW domains of YAP so that each domain can freely choose its more preferred PY-motif. The above structural analysis indicates that the YAP WW tandem would bind to the ligands containing two closely spaced PY motifs with only modest affinities due to the lack of large conformational coupling of its two WW domains. The WW tandems of Yorkie and YAP (Yorkie is the

YAP ortholog in *Drosophila*) bind to Dendrin PY23 with the comparable affinities (~500 nM vs ~400 nM, *Figure 1B* and *Figure 3—figure supplement 2E*). Howerver, the amino acid sequence linking the Yorkie WW tandem is totally different from that of YAP (*Figure 3—figure supplement 2F*), further indicating that the linker sequence connecting the YAP WW tandem does not play active structural roles during the protein evolution. In contrast, the inter-domain linker sequences of the KIBRA, MAGI2 and MAGI3 WW tandems are highly conserved during the evolution (*Figure 3—figure supplement 2A*).

## Origin of the exquisite target binding specificity of WW tandems

The similarities both in their structures and binding affinities to a set of common ligands indicate that the WW tandems of KIBRA, MAGI2 and MAGI3 share a common binding mode (*Figure 4A and B*, and *Figure 3—figure supplement 2C*). The extensive interactions between the inter-domain linker and the C-terminal helix couples the two WW domain into a structural supramodule in these WW tandems, thus providing a structural and thermodynamic basis for highly synergistic bindings to target proteins. For example, the formation of the WW tandem supramodule allowed KIBRA WW12 to bind to PTPN14 with an ~3900 fold higher affinity when compared to the isolated WW domains (*Figure 4D1* and *Figure 1—figure supplement 1B and C*). In contrast, the coupling between WW1 and WW2 in YAP is minimal (*Figure 4C*). Therefore, the target binding synergism between the two WW domains in the YAP tandem is also low. For example, the binding of YAP WW tandem to Dendrin PY23 is only a few tens tighter than the isolated WW domains (*Figure 4D2* and *Supplementary file 1B and D*), and this relatively low synergism likely originates from the very closely spaced PY-motifs of Dendrin PY23. Consistent with this analysis, very little synergism could be observed in the bindings of the YAP WW tandem to two PY motifs separated with extended linking sequences such as AMOT PY12 (*Figure 4D3* and *Supplementary file 1E*). NMR-based studies revealed that, in the absence of ligands, there is no direct conformational coupling between the two WW domains in the WW tandems of KIBRA (*Ji et al., 2019*), YAP (*Webb et al., 2011*), and MAGI2 (our unpublished data).

We directly tested the above WW domain coupling-induced target binding synergism model by experiments. In the WW tandem structures of KIBRA and MAGI2, a hydrophobic residue from WW1 (I35 in KIBRA and L330 in MAGI2) inserts into a hydrophobic pocket formed by WW2 domain and the inter-domain linker (*Figure 4A and B*). Substitution of this hydrophobic residue with a negatively charged Asp should impair this hydrophobic interaction and thus exert a negative impact on the synergism of the two WW tandems. Indeed, the I35D substitution weakened KIBRA's binding to PTPN14 PY12 by ~50 fold, and the L330D mutation decreased MAGI2's binding to Dendrin PY23 by ~57 fold (*Figure 4D*). L244 in YAP is the structurally equivalent residue of I35 in the KIBRA WW tandem (*Figure 4C*). Totally consistent with the structural finding of very little coupling between the two WW domains in YAP, the L244D mutation had no impact on YAP's binding to either Dendrin PY23 or AMOT PY12 (*Figure 4D*). The L244D mutation data of YAP WW12 further support that the mild synergism of the binding between the YAP WW tandem and Dendrin PY23 originates from the two closely spaced PY motifs of Dendrin.

The strong structural couplings between the two WW domains of the KIBRA and MAGI WW tandems indicate that their two PY-motif-binding pockets are fixed both in distance and orientation. Indeed, the distances between Cα atoms of the Trp residue at the end of β3 of WW1 (one of the two defining Trp residues in WW domains) and of the His residue in β2-β3 hairpin of WW2 (which is absolutely required for engaging the Tyr residue in the 'ΨPxY' motif) are in a very narrow range of 7.2–7.9 Å among the complex structures (*Figure 4E*). Correspondingly, the distances between the two PY motifs of the ligands, measured between the Cα of Tyr in the 1$^{st}$ PY and the Cα of Pro in 2$^{nd}$ PY, are also fixed at ~9.8 Å. The exception is the LATS1 PY23 motifs with the two Cα atoms separated by ~11.5 Å (*Figure 4E*). Notably, the PY-motifs in PTPN14, Dendrin and AMOT are all separated by only two residues, and all three ligands bind to the KIBRA, MAGI2 and MAGI3 WW tandems with very high affinities (*Figure 3—figure supplement 2B*). In contrast, the two PY-motifs of LATS1 are separated by three residues ('QGP') and LATS1 binds to the three WW tandems with quite modest affinities (*Figure 3D*). Strikingly, deleting the Gly residue (Gly553) in the three-residue linker of LATS1 PY23 converted the LATS1 mutant ('LATS1-Δ553') into a very strong binder of the KIBRA WW tandem ($K_d$ ~17 nM; *Figure 4F*), indicating that a two-residue linker between the two PY-motifs is optimal for specific and high affinity binding to the KIBRA, MAGI2 and MAGI3 WW

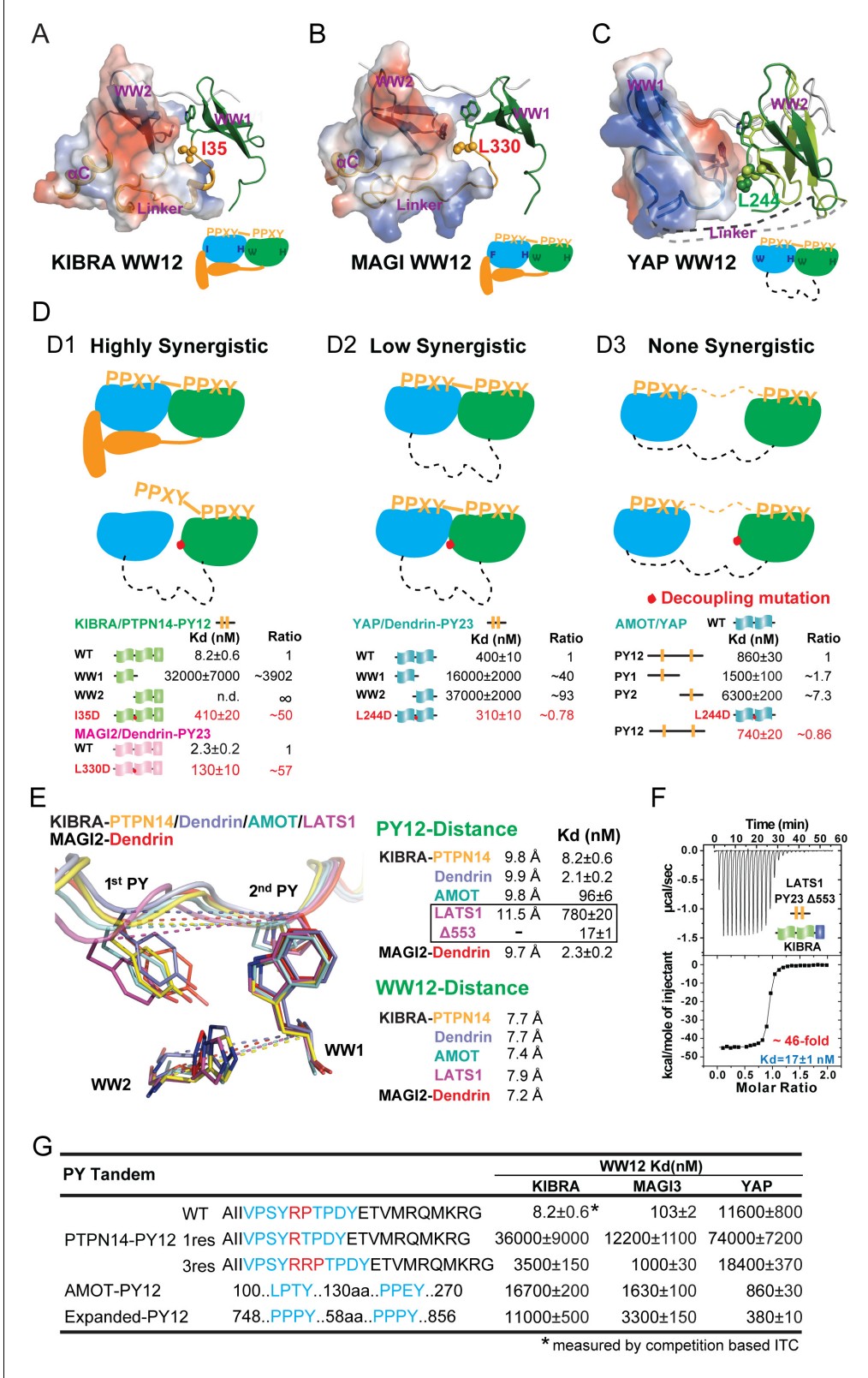

**Figure 4.** Binding modes of WW tandems to PY motifs. (**A–C**) Combined surface and ribbon diagram showing the domain coupling of the WW tandems of KIBRA (**A**), MAGI2 (**B**), and YAP (**C**), each in complex with Dendrin PY23. The cartoon in each panel is used to show the coupling mechanism of the WW domain tandems in the complexes. (**D**) Three distinct modes of WW tandem-mediated bindings to various PY motif targets. (**D1**), Highly synergistic bindings of tightly coupled WW domains in the tandem with two PY motifs separated by two and only two residues. In this mode, isolated

*Figure 4 continued on next page*

*Figure 4 continued*

WW domains have very weak binding to PY motifs as shown by the quantitative binding affinity data below the model cartoon. Perturbations of the inter-domain couplings (indicated by a red dot in the model corresponding to the I35D in KIBRA and L330D in MAGI2) also severely impair the WW tandem/target bindings. (**D2**) Low synergistic binding of WW tandems with PY motif targets as illustrated by the two WW domains with no apparent direct coupling but binding to two PY motifs separated only by a very short linker. In this mode, isolated WW domains have very weak binding to PY motifs. Modest synergism can be brought to action by the two closely spaced PY motifs. Changes of WW domain surface outside the binding areas (i.e. L244D of YAP) do not affect WW tandem/target bindings. (**D3**) None-synergistic bindings of WW tandems to PY motifs separated by long linkers. In this mode, bindings of the two WW domains to PY-targets are simply additive in their $K_d$ values. (**E**) Ribbon-stick model showing the distances of the two PY motifs as well as two WW domains in the five complex structures. The inter-WW domain distances are indicated by the signature Trp at the end of β3-strand of WW1 and His in the β2-β3 hairpin of WW2. The distances between the PY motifs are measured Tyr in the first PY motif and the first Pro in the second PY motif. The figure also shows that the $K_d$ values of the binds are nicely correlated to the distances between the two PY motifs in the complexes. (**F**) ITC-based measurement showing that removal of one residue (Gly553) in the 3-residue linker ('QGP') between LATS1 PY23 converted the LATS1 mutant into a super strong KIBRA WW tandem binder, presumably by shortening the inter-PY motif distance from 11.5 Å to ~9.8 Å as shown in panel E. (**G**) ITC-derived binding affinities of various WW tandems to PTPN14 PY12 motif mutants with different inter-motif linker lengths, showing that shortening or lengthening of the 2-residue PY motif linker led to weakening of the binding. AMOT PY12 and Expanded PY12 are used as examples of PY motifs separated by longer linkers.

DOI: https://doi.org/10.7554/eLife.49439.010

tandems. We further showed that shortening the linker of the PTPN14 PY12 to one residue (R, 1res) weakened its binding to both KIBRA and MAGI3 WW tandems by as much as ~4400 fold (*Figure 4G*; also see *Ji et al., 2019*). Similarly, lengthening the linker of the PTPN14 PY12 to three residues (RRP, 3res) also significantly weakened its bindings to the KIBRA and MAGI3 WW tandems (*Figure 4G*). In contrast, lengthening the linker of the PTPN14 PY12 to three residues had negligible impact on its binding to the YAP WW tandem, presumably its conformational flexibility allows the tandem to adjust the inter-domain distances and orientations.

In summary, we can divide the bindings between the WW domain tandems and two PY-motifs containing targets into three modes: 1), the highly synergistic and very strong binding mode, represented by the KIBRA, MAGI2 and MAGI3 WW tandems, which requires the tight conformational coupling of the two WW domains as well as two PY-motifs separated by two and only two residues (*Figure 4D1*); 2), the low synergistic binding mode with modestly high affinities, represented by the YAP WW tandem, which does not require tight coupling between the two WW domains but with two PY-motifs separated by a short linker of a few residues (*Figure 4D2*); and 3), those with no synergism and with weak binding affinities, in which the two WW domains do not couple with each other and the two PY motifs are separated by long linking sequences (*Figure 4D3*). The above findings are consistent with well-established thermodynamic principles governing multivalent protein/ligand interactions (see *Zhou and Gilson, 2009* for a review).

## The deletion of Gly553 in LATS1 inhibits YAP phosphorylation in cells

According to the biochemical and structural analysis, deleting Gly553 from the LATS1 PY23 converted LATS1-Δ553 into a strong binder of KIBRA. We confirmed that KIBRA indeed binds to LATS1-Δ553 stronger than to WT LATS1 using a co-immunoprecipitation-based assay with the full-length proteins expressed in HEK293A cells (*Figure 5A and B*). We further showed using purified recombinant proteins that KIBRA WW12 and YAP WW12 both formed complex with LATS1 PY1-4 on an analytical gel filtration column when the three proteins were mixed at a 1:1:1 ratio (*Figure 5—figure supplement 1A*), fitting with the observation that the two WW tandems have comparable affinities in binding to LATS1 PY1-4 (*Figure 1—figure supplement 1F*). In contrast, when YAP WW12, LATS1 PY1-4(Δ553) and KIBRA WW12 were mixed at a 1:1:1 ratio, essentially all KIBRA WW12 was found to be in complex with LATS1 PY1-4(Δ553) and YAP WW12 was found to be competed off and existing in the free form (*Figure 5—figure supplement 1B*). Interestingly, when Gly553 of LATS1 was substituted with Glu, the mutant LATS1 showed very weak binding to KIBRA (*Figure 5—figure supplement 1D*), and all KIBRA WW12 was found to be in the free form and most of YAP WW12 formed complex with the LATS1 mutant when KIBRA WW12, YAP WW12 and LATS1 PY1-4(G553E) were mixed at a 1:1:1 ratio (*Figure 5—figure supplement 1C*). The weakened binding of the LATS1-G553E mutant to KIBRA WW12 is likely due to more rigid backbone conformation of Glu over the

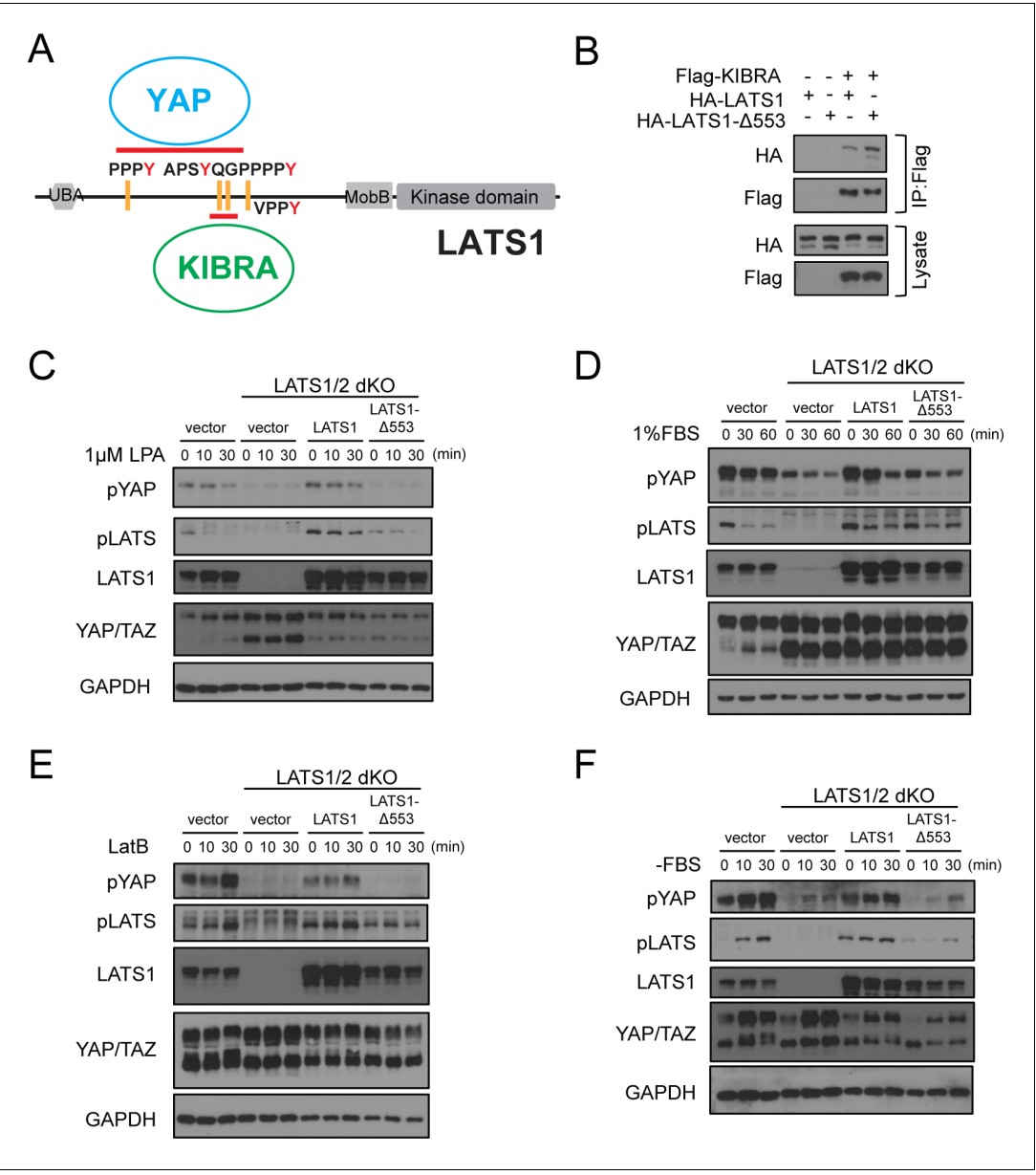

**Figure 5.** Deletion of Gly553 in LATS1 inhibits YAP phosphorylation in cells. (**A**) Schematic diagram showing the interaction of LATS with KIBRA or YAP. (**B**) LATS1-△553, compared to WT LATS1, has an enhanced binding to KIBRA in HEK293A cells. (**C, D**) Comparison of YAP and LATS1 phosphorylations of wild type HEK293A cells, LATS1/2 dKO cells, LATS1/2 dKO cells stably expressing LATS1 or LATS1-△553. Cell were serum-starved for 4 hr and then treated with 1 μM LPA (**C**) or 1%FBS (**D**) for indicated times. (**E, F**) Comparison of YAP and LATS1 phosphorylations of wild type HEK293A cells, LATS1/2 dKO cells, LATS1/2 dKO cells stably expressing LATS1 or LATS1-△553 treated with 0.2 mg/ml LatB (**E**) or serum starvation (**F**) for indicated times.
DOI: https://doi.org/10.7554/eLife.49439.011

The following figure supplement is available for figure 5:

**Figure supplement 1.** KIBRA and YAP compete for binding to LATS1.
DOI: https://doi.org/10.7554/eLife.49439.012

Glycine residue in the WT protein, such that the defined distance between the two PY-motif binding pockets of KIBRA WW12 can no longer accommodate the LATS1 PY23 mutant.

We speculated that the enhanced binding between KIBRA and LATS1-Δ553 would shift the LATS1 mutant from the YAP/LATS1 complex to the KIBRA/LATS1 complex and thus may diminish

the LATS1-induced YAP phosphorylation when cells are stimulated with the Hippo pathway activation signals. To test this hypothesis, we used the LATS1/2 dKO HEK293A cells that we characterized earlier (*Plouffe et al., 2016*). We generated stable cell pools by re-expressing WT LATS1 or LATS1-Δ553 in the LATS1/2 dKO cells to test YAP and LATS1 phosphorylations upon various Hippo pathway stimulation signals. Upon treatment with lysophosphatidic acid (1 µM LPA), serum stimulation (1% FBS), latrunculin B (LatB, 0.2 mg/ml), or serum starvation (-FBS), which are well-known signals for the Hippo pathway (*Yu et al., 2012*; *Meng et al., 2015*), cells expressing LATS1-Δ553 had diminished YAP and LATS1 phosphorylations when compared with cells expressing WT LATS1 (*Figure 5C–F*). These data suggest that the interaction of LATS1 with tandem WW domain-containing proteins, such as KIBRA, may play a role in Hippo pathway regulation.

## Further affinity determinants of PY-motifs for highly synergistic bindings to WW tandems

Several additional sequence features (denoted as 'Enhancers' in *Figure 6A*) contribute to the very high affinity bindings to the tightly coupled WW tandems. First, residues in the two-residue PY-motif linker can enhance binding. For example, R440 of PTPN14 PY12 forms salt bridges with D17 and D19 in the WW1 of the KIBRA WW tandem (*Figure 2A*). Substitution of R440 with Ala weakened PTPN14's binding to KIBRA by 17-fold (*Figure 6A and B*). Second, the C-terminal extension of PTPN14 PY12 tandem enhances its binding to KIBRA WW tandem by forming hydrophobic interactions with its WW1 domain (*Figure 6A*). Truncating this C-terminal extension weakened PTPN14 PY12's binding to the WW tandems of KIBRA and MAGI3 ('Del-C' in *Figure 6B*). Third, converting the non-canonical PY-motifs into canonical ones (i.e. 'PPxY') further enhanced PTPN14 PY12's binding to both KIBRA and MAGI3 WW tandems ('2Pro' and '4Pro' in *Figure 6B*). Finally, although not tested here, one may be able to further enhance the ligand binding to the KIBRA and MAGI WW tandems by utilizing a small unoccupied hydrophobic surface of WW2 to bind to residue(s) N-terminal to the first PY-motif (the dashed oval in *Figure 6A*) (*Qi et al., 2014*; *Liu et al., 2016*).

The finding that the canonical WW domains can bind to non-canonical PY-motif sequences significantly expands possible WW domain/target binding space. It is thus important to define which residues are allowed in the first position of the 'ΨPxY' motif in addition to Pro for binding to the canonical WW domains. We modified the first Pro of the second PY in the peptide 'DRPPPYVA<u>P</u>-<u>PSY</u>EG' (the bold face 'P' in the Dendrin PY23 peptide without the C-terminal extension) with Leu, Ala, Val, Cys, Ser, Ile, Thr, and Met, and measured the bindings of these peptides to the KIBRA and MAGI3 WW tandems by ITC (*Figure 6C* and *Figure 6—figure supplement 1*). As expected, all substitutions weakened the peptide's binding to the tandems (*Figure 6C,D and E*, and *Figure 6—figure supplement 1*). However, substitutions of the Pro residue with Cys, Ala, Ser, Thr, and Val lead to only mild weakening of the binding, whereas substitutions with bulky hydrophobic Leu, Met and Ile essentially disrupted the binding. The above results can be rationalized by the structure of the KIBRA WW tandem in complex with PTPN14 PY12 showing that the pocket accommodating the residue (T442 in *Figure 6A*) is quite small.

Our structural studies of the five WW tandem/target complexes also revealed that the sequence requirement for the first PY motif is relatively loose. Though the 'PPxY' sequence is optimal (*Figure 6B*), the two Pro residues can be other aliphatic residues or even uncharged polar amino acids (*Figure 2*). This can be explained by the relatively shallow and open hydrophobic pocket that the WW2 domain uses to accommodate the first two residues from the first PY motif (e.g. V436 and P437 in *Figure 6A*). Taken all above results together, we can deduce a consensus sequence motif for specific and strong bindings to the WW-tandems of KIBRA, MAGI2, and MAGI3 as: 'ΦΦxYxxΨPxY', where Φ is aliphatic or small polar amino acids and Ψ can be Pro, Cys, Ala, Ser, Thr or Val (*Figure 7A*). The first PY-motif of such sequence binds to WW2 and the second PY-motif binds to WW1 of the WW tandems (*Figure 7B*).

## Identification of new binders of the KIBRA, MAGI2, and MAGI3 WW tandems

We searched for proteins in the UniProtKB/Swiss-Prot database (*UniProt Consortium, 2019*) and identified 132 human proteins containing the 'ΦΦxYxxΨPxY' motifs (*Supplementary file 3*). Besides Dendrin, only two proteins, junctional protein associated with coronary artery disease (JCAD) and

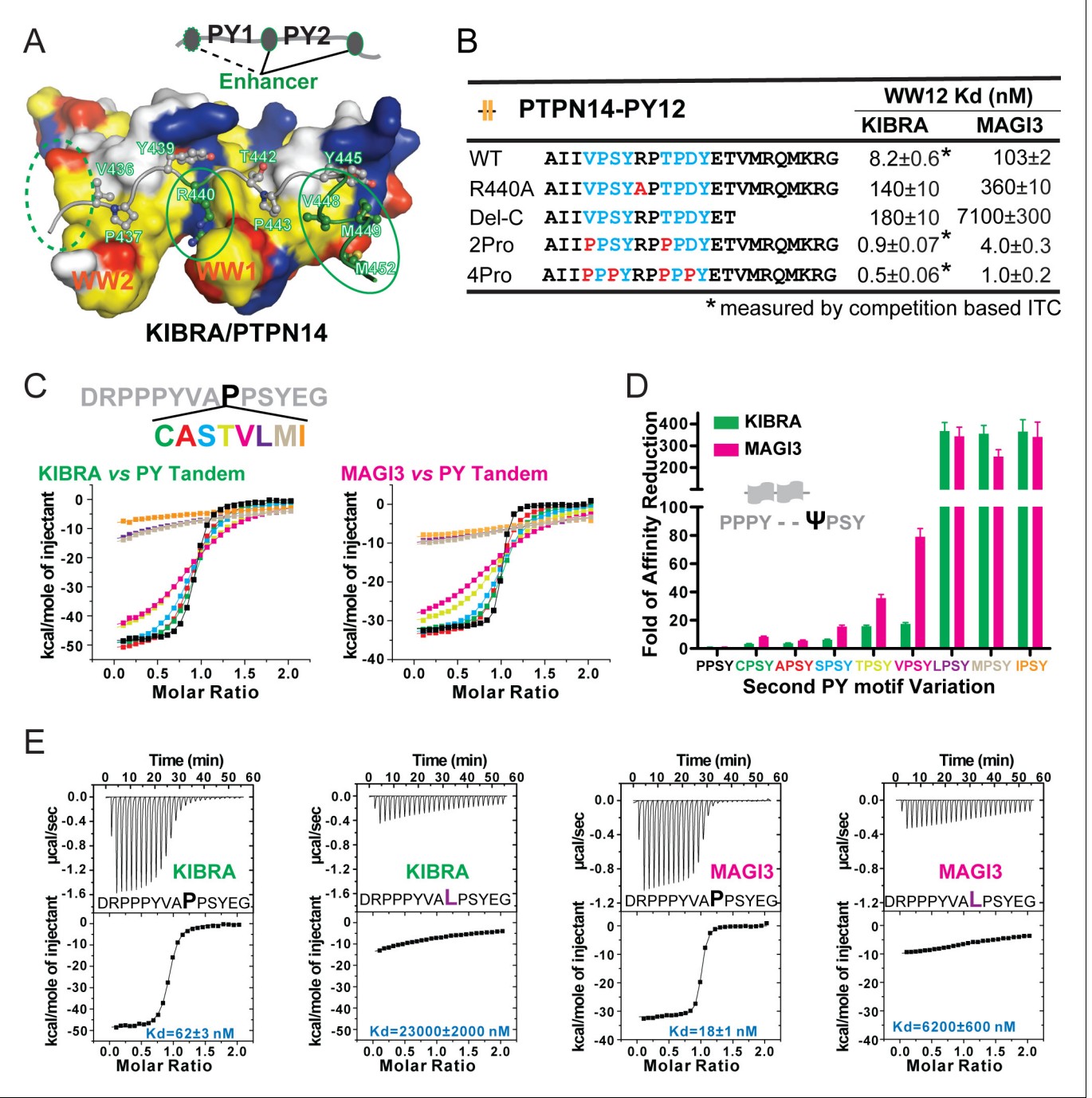

**Figure 6.** Specificity determinants of closely spaced PY motif to the WW tandems of KIBRA and MAGI2/3. (**A**) Surface combined with ribbon-stick model showing the binding between the KIBRA WW tandem and PTPN14 PY12. The figure is to show that, in addition to the WW/PY motif interactions, the inter-PY motif linker and the extension sequence following PY2 (as indicated by the green ovals) also play critical roles for the super strong bindings between the WW tandem and the PY motifs. Additionally, residues preceding PY1 (indicated by the dotted green oval) may also enhance the binding. In the drawing, the hydrophobic, negatively charge, positively charged, and polar residues are colored in yellow, red, blue and white, respectively, in the surface model. (**B**) ITC-derived binding affinities of the KIBRA and MAGI3 WW tandem to various PTPN14 PY12 mutants. For bindings with nM or sub-nM $K_d$ values, competition-based ITC assays were used as described in *Figure 1C and D*. (**C**) Mutational analysis of the contributions of different amino acids corresponding to the first Pro in the second PY motif of Dendrin PY23 in binding to KIBRA and MAGI3 WW tandems. The figure shows the superposition plot of ITC-based binding curves of binding reactions performed under the same experimental condition (also see *Figure 6—figure supplement 1* for re-scaled ITC curve) (**D**) Comparison of relative binding affinity difference of various PY motif mutants tested in Panel C. In this comparison, the bindings of the KIBRA and MAGI3 WW tandems to the wild type PY motifs with the optimal 'PPSY' sequence are set at the base value

*Figure 6 continued on next page*

*Figure 6 continued*

of '1'. The fold of affinity reductions of each mutant are plotted. (**E**) The representative ITC curves of showing the binding of the KIBRA/MAGI3 WW tandems to the WT PY motif peptide or to one of the representative mutants.

DOI: https://doi.org/10.7554/eLife.49439.013

The following figure supplement is available for figure 6:

**Figure supplement 1.** Bindings between WW tandems and closely spaced PY motifs with 2[nd] PY variation.

DOI: https://doi.org/10.7554/eLife.49439.014

USP6 N-terminal-like protein (USP6NL), contain two canonical 'PPxY' motifs separated by exactly two residues (*Figure 7C*). PTPN21 contains two consensus sequences with its PY12 highly similar to PTPN14 PY12 and PY34 being unique to PTPN21 itself (*Figure 7C*).

We selected 7 of such consensus sequences from six proteins (JCAD, USP6NL, β-Dystroglycan, PTPN21, ABLIM-1, and PTCH1) to measure their bindings to the KIBRA and MAGI3 WW tandems by ITC (*Figure 7C* and *Figure 7—figure supplement 1*). As expected, both JCAD and USP6NL bind strongly to the KIBRA WW tandem (*Figure 7C and D*). Both PY12 and PY34 of PTPN21 show strong bindings to the KIBRA WW tandem. PTPN21 PY12 and PTPN14 PY12 bind to KIBRA WW tandem with essentially the same affinity ($K_d$ ~8 nM), as the sequences of these two peptides are very similar. Interestingly, β-Dystroglycan, a protein known to play critical roles in Hippo signaling-related cell adhesion and cell growth (*Morikawa et al., 2017*; *Gawor and Prószyński, 2018*), can also bind to KIBRA WW tandem with a high affinity ($K_d$ ~96 nM, *Figure 7E*). We solved the crystal structure of the KIBRA WW tandem in complex with the β-Dystroglycan PY34 peptide. The complex structure confirmed that the peptide binds to the KIBRA WW tandem following the mode exactly as we have predicted (*Figure 7F*). Therefore, β-Dystroglycan may be a key regulator in cell growth and neuronal synaptic functions via specifically binding to KIBRA.

## Discussion

In this study, we discovered that WW tandems, but not individual WW domains, can bind to target proteins with extremely high affinity and with very high sequence specificity. Such strong bindings are realized by both highly synergistic conformational coupling of two WW domains connected in tandem and two PY-motifs separated by a two and only two amino acid residues linker. If only using either one of the above two elements, the bindings become to be weakly synergistic and with modest affinities with $K_d$ of a few hundreds of nM to a few µM. When two uncoupled WW domains binding to two PY motifs separated by more than two residues, their bindings are weak with $K_d$ of a few to a few tens µM or even weaker due to very little or no synergism of the bindings between the two domains (*Figure 4D*). Weak targeting binding synergisms resulted from week couplings of WW domains connected in tandem have also been reported in several other cases in the literature (*Aragón et al., 2011*; *Verma et al., 2018*; *Fedoroff et al., 2004*; *Liu et al., 2016*; *Qi et al., 2014*). Additionally, it is well known that isolated WW domains bind to their targets with low affinities ($K_d$ in the range of a few tens to a few hundreds of µM) (*Sudol and Hunter, 2000*; *Salah, 2012*). Therefore, WW domains can bind to their target proteins with a very broad affinity range with $K_d$ values range from a few nM to hundreds of µM or weaker. Presumably, those high affinity WW tandem-mediated target interactions should play specific functions. For example, we recently showed that the tight binding between KIBRA and Dendrin ($K_d$ ~2.1 nM) is critical for KIBRA (and its associated AMPA receptors) localization in neuronal synapses and for synaptic plasticity (*Ji et al., 2019*). We have also demonstrated that bindings of short PY motif containing peptides to WW tandems can be readily enhanced to sub-nanomolar affinities (*Figure 6B*), and these super-strong WW tandem binding peptides may be used as effective tools for studying the role of specific interactions between WW tandem proteins and target proteins (see *Ji et al., 2019* for an example). As for the very weak WW domain/target interactions, cautions are required in interpreting potential functional implications of such bindings.

The enhanced PY-motif containing target binding afforded by WW domains connected in tendem is in line with the Second Law of Thermodynamics. For a tandem/bivalent system, the binding energy can be written as $\Delta G_{total} = \Delta G_1 + \Delta G_2 - \Delta G_c$; where $\Delta G_1$ and $\Delta G_2$ represent the binding free energy of the two individual sites and $\Delta G_c$ can be viewed as amount of energy penalty (or conformational

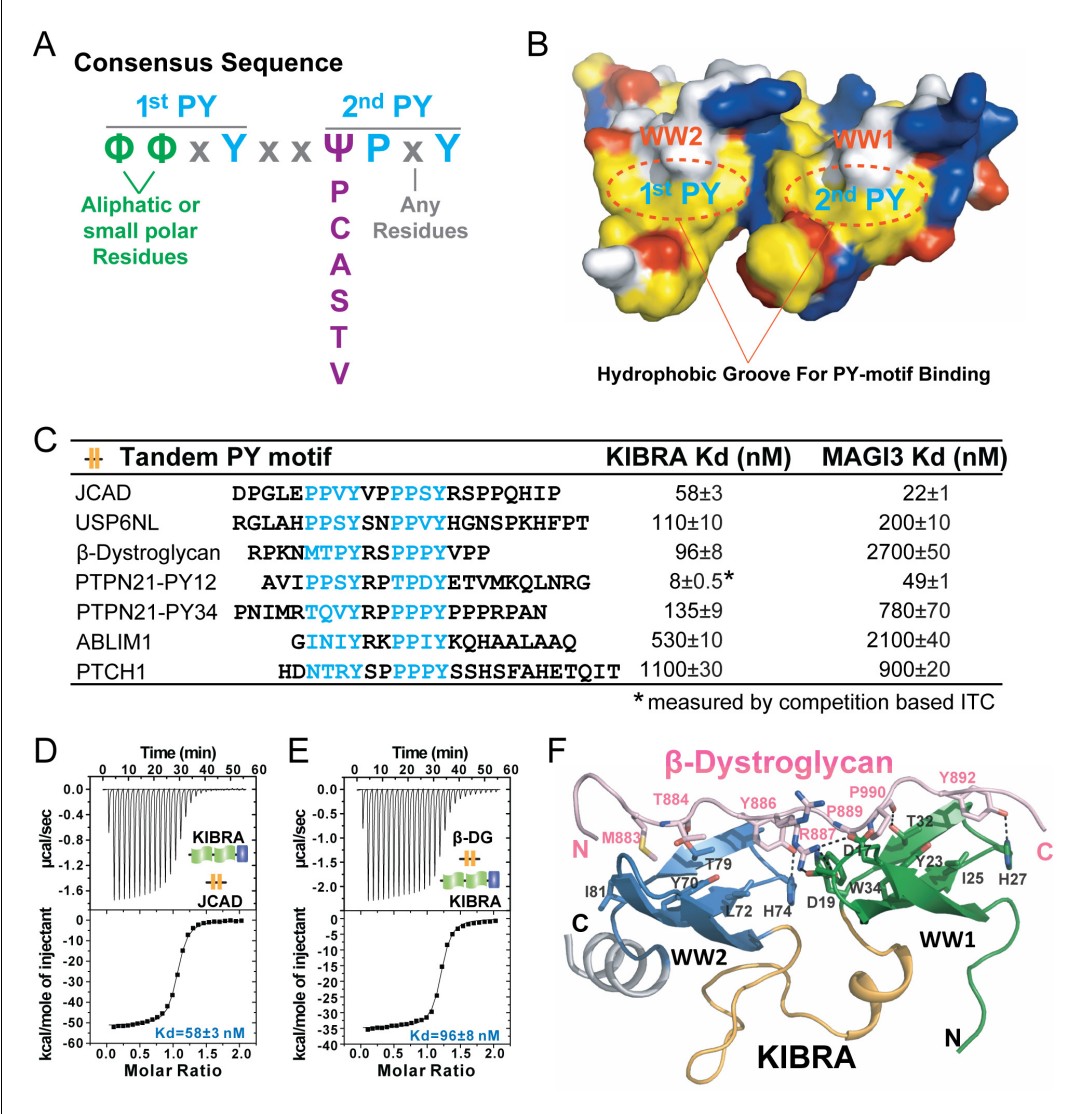

**Figure 7.** A consensus sequence of closely-spaced PY motifs capable of binding to KIBRA, MAGI2, and MAGI3 WW tandems with high affinities. (**A**) The consensus sequence pattern of two PY motif-containing sequences that can synergistically bind to tightly coupled WW tandems such as those from KIBRA, MAGI2, and MAGI3. (**B**) Surface diagram showing two hydrophobic PY motif binding pockets of WW tandem that are precisely spaced due to the tight inter-domain coupling. The color coding is the same as in *Figure 5A*. (**C**) ITC-based measurements of the binding affinities of selected targets with PY motifs fitting the consensus sequence indicated in panel A to WW tandems from KIBRA and MAGI3. (**D and E**) Representative ITC binding curves of the bindings described in panel C, showing the bindings of KIBRA WW tandem to PY motifs JCAD (**D**) and β-Dystroglycan (**E**). (**F**) Ribbon-stick model showing the structure and detailed interaction of the KIBRA WW tandem in complex with the PY motifs from β-Dystroglycan.

DOI: https://doi.org/10.7554/eLife.49439.015

The following figure supplement is available for figure 7:

**Figure supplement 1.** Domain organization of selected proteins with PY motifs fitting the consensus sequence indicated in panel *Figure 7A*.

DOI: https://doi.org/10.7554/eLife.49439.016

entropy loss) due to non-perfect conformational coupling of the two binding sites (*Zhou and Gilson, 2009*). For a system with perfect conformational coupling, $\Delta G_c$ would be approaching zero. For a system without any conformational coupling, the two bind sites can be viewed as completely separate (i.e. the stronger binding site prevails). $\Delta G_c$ often falls in between the above two extreme conditions in real biological systems. For the KIBRA WW tandem, the situation is even more complicated as its WW2 is only partially folded in the absence of ligands (*Ji et al., 2019*). The total free energy will also include a binding-induced domain folding energy.

As for the WW domain-mediated protein-protein interactions in cell growth and polarity, the results presented in the current study will be valuable to re-evaluate many of reported protein-protein interactions and for guiding experimental designs in the future. For example, the bindings of the KIBRA, MAGI2, and MAGI3 tandems to PTPN14 (and PTPN21) are so much stronger than the binding of YAP to PTPN14 (~1,400 fold difference; *Figure 1B*). Therefore, it is almost for sure that PTPN14 should first bind to KIBRA and MAGI instead of to YAP when PTPN14 is not in excess. The binding between PTPN14 to YAP can occur if PTPN14 is in excess of the total amount of KIBRA and MAGI together. Alternatively, PTPN14 may use its PY34 to bind to the YAP WW tandem via rather weaker binding (*Wang et al., 2012*; *Huang et al., 2013*; *Liu et al., 2013*; *Michaloglou et al., 2013*). Similarly, the binding of the KIBRA tandem (or MAGI2 and MAGI3 WW tandems) to LATS1 is ~4.5 fold stronger than YAP tandem to LATS1 (*Figure 3D* and *Figure 1—figure supplement 1F*), therefore KIBRA might be able to modulate the concentration of the LATS1/YAP complex formation by competing with YAP for binding to LATS1. This is supported by the finding that further enhancing the KIBRA/LATS1 binding by deleting Gly553 in the LATS1 PY23 linker can effectively block LATS1-mediated YAP phosphorylation, presumably due to the effective sequestration of LATS1-Δ553 by KIBRA (*Figure 5*). It should be noted that there are multiple strong KIBRA WW tandem binders such as PTPN14, PTPN21, Dendrin, AMOT, β-Dystroglycan, and JCAD (*Figure 1B* and *Figure 7C*), these molecules can tune the availability of free KIBRA in competing with YAP for binding to LATS. One may view that proteins like KIBRA and MAGIs function as an upstream nexus that can sense various cell growth and polarity cues and then determine whether and how much the LATS/YAP complex can form (i.e. determining the level of YAP phosphorylation by LATS) in living tissues. Conversely, certain PY motif containing proteins such as AMOTs can effectively compete with LATS for binding to YAP (*Figure 1E*) and thereby regulate the LATS/YAP complex formation. It is noted that most of the studies in elucidating molecular roles of WW domain proteins and their PY motif-containing targets involved in cell growth and cell polarity in the literature used protein overexpression or removal approaches. Such methods unavoidably would alter the WW domain-mediated protein-protein interaction network, which is very sensitive to the concentrations of each protein based on the quantitatively binding affinity data presented in this work. Therefore, we advocate that, when studying proteins in cell growth and polarity, cares should be taken to consider the quantitative binding affinities and endogenous concentrations of these proteins. Finally, our study also indicates that mutations of either WW tandem proteins or their PY-motif containing targets found in patients not only can affect the bindings of their direct targets, but may also alter other protein interactions in this intricate cell growth and polarity regulatory protein network organized by the WW tandem/PY-motif interactions.

## Materials and methods

All reagents and analytical tools used for this paper and described in the methods below are compiled as a table in the *Supplementary file 4*.

### Constructs, Protein Expression and Purification

The coding sequences of desired constructs were PCR amplified from mouse brain cDNA libraries. For crystallization, Dendrin (222-241) was covalently linked at N- or C-terminal of YAP (156-247) by a two-step PCR-based method (referred to as Dendrin-'L'-YAP and YAP-'L'-Dendrin). All site mutations were created by PCR and confirmed by DNA sequencing. All constructs used for protein expression were individually cloned into a pET vector and recombinant proteins were expressed in Codon-plus BL21 (DE3) *Escherichia coli* cells with induction by 0.3 mM IPTG at 16°C overnight. All recombinant proteins were purified using $Ni^{2+}$-nitrilotriacetic acid agarose (Ni-NTA) column followed by size-exclusion chromatography (Superdex 200 column from GE Healthcare) in a final buffer containing 50 mM Tris-HCl (pH 7.8), 100 mM NaCl, 1 mM DTT and 1 mM EDTA. The selenomethionine-labeled proteins were expressed by methionine auxotroph *E. coli* B834 cells in LeMaster media and was purified following the same protocol used for the wild-type proteins. All tags of recombinant proteins were removed before crystallization, except for the $His_6$-tagged YAP-'L'-Dendrin.

## Isothermal Titration Calorimetry assay

Isothermal Titration Calorimetry (ITC) experiments were carried out on a VP-ITC calorimeter (Micro-Cal) at 25°C. Titration buffer contained 50 mM Tris-HCl (pH 7.8), 100 mM NaCl, 1 mM DTT and 1 mM EDTA. For a typical experiment, each titration point was performed by injecting a 10 μL aliquot of protein sample (50–300 μM) into the cell containing another reactant (5–30 μM) at a time interval of 120 s to ensure that the titration peak returned to the baseline. For the competition experiments, proteins in the syringe were titrated to a mixture with a 2-fold molar concentration excess of a competitor over the reactants in the cell. For the ITC-based binding assays, most of the PY-motif containing peptide fragments were prepared by fusing each peptide fragment to the C-terminal end of the TRX-tag (see *Supplementary file 4* for details). Several cases were chosen to investigate whether the Trx fusion tag might impose certain impact on the PY motif peptides in their bindings to WW domains. We chose several minimal PY motif peptides (e.g. the Dendrin PY23 and β-DG PY34) for such analysis. It was reasoned if the fusion tag does not affect the minimal PY-motif peptides in binding to their targets, the same fusion tag likely would not alter the bindings of longer PY motif-containing peptides with more flexible residues surrounding the PY motif to WW domains. We showed that the TRX-tag had a negligible impact on the bindings of the Dendrin PY23 peptide and the β-DG PY34 peptide to KIBRA WW12 (*Figure 1—figure supplement 2*). The titration data were analyzed with Origin7.0 (MicroCal) using a one-site binding or competitive binding model.

## Crystallization and data collection

All crystals were obtained by hanging drop vapor diffusion methods at 4°C or 16°C within 3–5 days. Crystals of wild-type and Se-Met-substituted KIBRA/LATS1 were grown in solution containing 20–30% w/v PEG 4000 and 100 mM Tris (pH 8.0); crystals of KIBRA/AMOT were grown in solution containing 20–25% w/v pentaerythritol propoxylate 629 (17/8 PO/OH), 50 mM $MgCl_2$ and 100 mM Tris (pH 8.5); crystals of KIBRA/PTPN14 were grown in solution containing 700 mM magnesium formate and 100 mM Bis-Tris Propane (pH 7.0); crystals of KIBRA/β-Dystroglycan were grown in solution containing 3.0–4.0 M NaCl and 100 mM Bis-Tris (pH 6.5); crystals of MAGI2/Dendrin were grown in solution containing 2.0–2.5 M ammonium sulfate and 100 mM sodium acetate (pH 4.6); crystals of Dendrin-'L'-YAP were grown in solution containing 28–35% w/v pentaerythritol propoxylate 426 (5/4 PO/OH) and 100 mM HEPES (pH 7.5); and crystals of $His_6$-tagged YAP-'L'-Dendrin were grown in solution containing 25–30% w/v PEG400, 200 mM $CaCl_2$ and 100 mM HEPS (pH 7.5). Before diffraction experiments, crystals were soaked in the original crystallization solutions containing an additional 10–25% glycerol for cryoprotection. All diffraction data were collected at the Shanghai Synchrotron Radiation Facility BL17U1 or BL19U1. Data were processed and scaled by HEL2000 or HKL3000.

## Structure determination and refinement

For the KIBRA/LATS1 complex, the program HKL2MAP (*Pape and Schneider, 2004*) yielded two Se sites in one asymmetric unit. The initial SAD phases were calculated using PHENIX (*Adams et al., 2010*). Other structures were determined by molecular replacement using PHASER (*McCoy et al., 2007*). The initial phases of KIBRA/ligands complex were solved using the WW tandem part from the structure of KIBRA/LATS1 complex as the searching model. The initial phase of MAGI2/Dendrin complex was solved using the structures of MAGI1-WW1 (PDB: 2YSD) and MAGI1-WW2 (PDB: 2YSE) as the searching models. The initial phases of $His_6$-tagged YAP-'L'-Dendrin was solved using the structures of YAP-WW1 (PDB: 4REX) and YAP-WW2 (PDB: 2L4J) as the searching models. The initial phases of Dendrin-'L'-YAP was solved using the structures of YAP-WW1/2 from the structure of YAP-'L'-Dendrin. PY tandems were manually built according to the $F_{obs}$-$F_{calc}$ difference Fourier maps in COOT (*Emsley et al., 2010*). Further manual model adjustment and refinement were completed iteratively using COOT (*Emsley et al., 2010*) and PHENIX (*Adams et al., 2010*) based on the $2F_{obs}$-$F_{calc}$ and $F_{obs}$-$F_{calc}$ difference Fourier maps. The final models were further validated by MolProbity (*Williams et al., 2018*). Detailed data collection and refinement statistics are summarized in *Supplementary file 1*. All structure figures were prepared using PyMOL (http://pymol.sourceforge.net/).

## Analytical gel filtration chromatography

Analytical gel filtration chromatography was carried out on an AKTA FPLC system (GE Healthcare). Protein samples (each protein at a concentration of 40 µM) were loaded to a Superose 12 10/300 GL column (GE Healthcare) pre-equilibrated with assay buffer (same with ITC).

## Cell culture, transfection and retroviral infection

All cell lines were maintained in DMEM (GIBCO) containing 10% fetal bovine serum (FBS, GIBCO), 100 U/mL penicillin and 100 µg/mL streptomycin and with 5% $CO_2$ at 37 ˚C. To stimulate the Hippo pathway, low confluence cells ($1.0 \times 10^5$ cells per well) in 12-well plates were treated with serum stimulation, serum starvation, LPA (1 µM), and LatB (0.2 mg/ml). HEK293A cell line was from Dr. Kun-Liang Guan's lab (originated from ATCC). Cells were tested using morphology, karyotyping, and PCR-based approaches to confirm their identity. Cells were tested negative for mycoplasma contamination by cytoplasmic DAPI staining.

The human LATS1-Δ553 plasmid was cloned by PCR and confirmed by DNA sequencing. Cells were seeded in 6-well plates for 24 hr, and then were transfected with plasmids using PolyJet Reagent (Signagen Laboratories) according to manufacturer's instruction. To generate HEK293A cells stably expressing WT LATS1 or LATS1-△553, stable HEK293A cells with LATS1/2 dKO (*Plouffe et al., 2016*) were individually infected with retrovirus packed with plasmid expressing empty vector (pQCXIH), LATS1, or LATS1-△553. Forty-eight hours after infection, cells were selected with 250 µg/mL hygromycin (Roche) in the culture medium.

## Immunoblotting and immunoprecipitation

Immunoblotting was performed with the standard methods as described (*Yu et al., 2012*; *Meng et al., 2015*). Antibodies for pYAP(S127) (#4911), LATS1(#3477), pLATS(#8654S), HA-HRP (#2999) were from Cell Signaling Technology; Anti-Flag antibody (#F1804) was from Sigma-Aldrich; for YAP/TAZ(#sc-101199) was from Santa Cruz Biotechnology.

For immunoprecipitaion, cells were harvested at 48 hr after transfection with lysis buffer containing 50 mM Tris (pH 7.5), 150 mM NaCl, 1 mM EDTA, 1% NP-40, 10 mM pyrophosphate, 10 mM glycerophosphate, 50 mM NaF, 1.5 mM $Na_3VO_4$, protease inhibitor cocktail (Roche), and 1 mM PMSF. After centrifuging at 13,300 × g for 15 min at 4˚C, supernatants were collected for immunoprecipitation. Anti-flag antibody was added to the supernatants and the mixtures were incubated overnight at 4˚C. Then Pierce Protein A/G Magnetic Beads were added to the mixtures and further incubated for 2 hr at 4˚C. Immunoprecipitate proteins were eluted with SDS-PAGE sample buffer, reolved by SDS-PAGE and probed by the anti-HA antibody.

## Data resources

The atomic coordinates of the WW tandem and target complex structures have been deposited to the Protein Data Bank under the accession codes of: 6J68 (KIBRA/LATS1), 6JJW (KIBRA/PTPN14), 6JJX (KBIRA/AMOT), 6JJY (KIBRA/β-DG), 6JJZ (MAGI2/Dendrin), 6JK0 (YAP-Linker-Dendrin), and 6JK1 (Dendrin-Linker-YAP).

## Acknowledgements

We thank the BL19U1 beamline at National Facility for Protein Science Shanghai (NFPS) and BL17U1 beamline at Shanghai Synchrotron Radiation Facility (SSRF) for X-ray beam time. This work was supported by grants from RGC of Hong Kong (AoE-M09-12 and C6004-17G), a grant from Asia Foundation for Cancer Research (AFCR17SC01) to MZ, and grants from National Institute of Health (CA196878, CA217642, GM51586, DEO15964) to KLG. MZ is a Kerry Holdings Professor in Science and a Senior Fellow of IAS at HKUST.

## Additional information

### Competing interests
Mingjie Zhang: Reviewing editor, *eLife*. Kunliang Guan: co-founder and has an equity interest in Vivace Therapeutics, Inc. The terms of this arrangement have been reviewed and approved by the University of California, San Diego in accordance with its conflict of interest policies. The other authors declare that no competing interests exist.

### Funding

| Funder | Grant reference number | Author |
|---|---|---|
| Asia Foundation for Cancer Research | AFCR17SC01 | Mingjie Zhang |
| National Institutes of Health | CA196878 | Kunliang Guan |
| Research Grants Council, University Grants Committee | AOE-M09-12 | Mingjie Zhang |
| Research Grants Council, University Grants Committee | C6004-17G | Mingjie Zhang |
| National Institutes of Health | CA217642 | Kunliang Guan |
| National Institutes of Health | GM51586 | Kunliang Guan |
| National Institutes of Health | DEO15964 | Kunliang Guan |

The funders had no role in study design, data collection and interpretation, or the decision to submit the work for publication.

### Author contributions
Zhijie Lin, Formal analysis, Validation, Investigation, Writing—original draft, Writing—review and editing; Zhou Yang, Formal analysis, Investigation, Methodology, Writing—original draft; Ruiling Xie, Zeyang Ji, Formal analysis, Investigation; Kunliang Guan, Formal analysis, Supervision, Methodology, Writing—review and editing; Mingjie Zhang, Conceptualization, Resources, Supervision, Writing—original draft, Project administration, Writing—review and editing

### Author ORCIDs
Ruiling Xie ⓘ http://orcid.org/0000-0002-6086-8683
Mingjie Zhang ⓘ https://orcid.org/0000-0001-9404-0190

### Decision letter and Author response
Decision letter https://doi.org/10.7554/eLife.49439.037
Author response https://doi.org/10.7554/eLife.49439.038

## Additional files

### Supplementary files
• Supplementary file 1. Statistics of Data Collection and Model Refinement of the crystal structures.
DOI: https://doi.org/10.7554/eLife.49439.017

• Supplementary file 2. Summary of thermodynamic parameters for the binding of each WW domain or tandem to motifs derived by ITC experiments.
DOI: https://doi.org/10.7554/eLife.49439.018

• Supplementary file 3. List of human proteins containing the 'ΦΦxYxxΨPxY' motif.
DOI: https://doi.org/10.7554/eLife.49439.019

• Supplementary file 4. Key Resources Table.
DOI: https://doi.org/10.7554/eLife.49439.020

• Transparent reporting form

DOI: https://doi.org/10.7554/eLife.49439.021

## Data availability

The atomic coordinates of the WW tandem and target complex structures have been deposited to the Protein Data Bank under the accession codes of: 6J68 (KIBRA/LATS1), 6JJW (KIBRA/PTPN14), 6JJX (KBIRA/AMOT), 6JJY (KIBRA/β-DG), 6JJZ (MAGI2/Dendrin), 6JK0 (YAP-Linker-Dendrin), and 6JK1 (Dendrin-Linker-YAP).

The following datasets were generated:

| Author(s) | Year | Dataset title | Dataset URL | Database and Identifier |
|---|---|---|---|---|
| Lin Z, Yang Z, Zhang M | 2019 | PDB coordinates KIBRA/LATS1 | http://www.rcsb.org/structure/6J68 | RCSB Protein Data Bank, 6J68 |
| Lin Z, Yang Z, Zhang M | 2019 | PDB coordinates KIBRA/PTPN14 | http://www.rcsb.org/structure/6JJW | RCSB Protein Data Bank, 6JJW |
| Lin Z, Yang Z, Zhang M | 2019 | PDB coordinates KBIRA/AMOT | http://www.rcsb.org/structure/6JJX | RCSB Protein Data Bank, 6JJX |
| Lin Z, Yang Z, Zhang M | 2019 | PDB coordinates KIBRA/$\beta$-DG | http://www.rcsb.org/structure/6JJY | RCSB Protein Data Bank, 6JJY |
| Lin Z, Yang Z, Zhang M | 2019 | PDB coordinates MAGI2/Dendrin | http://www.rcsb.org/structure/6JJZ | RCSB Protein Data Bank, 6JJZ |
| Lin Z, Yang Z, Zhang M | 2019 | PDB coordinates YAP-Linker-Dendrin | http://www.rcsb.org/structure/6JK0 | RCSB Protein Data Bank, 6JK0 |
| Lin Z, Yang Z, Zhang M | 2019 | PDB coordinates Dendrin-Linker-YAP | http://www.rcsb.org/structure/6JK1 | RCSB Protein Data Bank, 6JK1 |

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
