## [Decision Letter]

Thank you for submitting your article "Decoding WW Domain Tandem-mediated Target Recognitions in Tissue Growth and Cell Polarity" for consideration by *eLife*. Your article has been reviewed by three peer reviewers, including William I Weis as the Reviewing Editor and Reviewer #1, and the evaluation has been overseen by a Reviewing Editor and John Kuriyan as the Senior Editor.

The reviewers have discussed the reviews with one another and the Reviewing Editor has drafted this decision to help you prepare a revised submission.

Summary:

This is a thorough and rigorous study of the interactions of tandem WW domains with tandem peptide ligands. Although the binding of WW domains to their ligands has been characterized previously, the affinities are low, making it hard to discriminate among potential PY motif-containing ligands. The authors demonstrate that a specific structural arrangement of tandem WW domains can produce to high-affinity binding. The enhancement is tuned by the spacing, rigidity and sequence of the linker, and the tandem also enables binding to non-canonical sequences as well as sequences flanking the PY motifs. An important result is that the specific tandem arrangement of WW domains can relax the canonical "ΨPxY" motif in the ligand that binds to the second WW domain and enable specific interactions with residues that flank the motif. Searching this relaxed, extended motif throughout the human proteome led to the identification of novel high-affinity binders to MAGI3 and KIBRA. This detailed study will have a major impact on how researchers think about WW domain protein interaction networks in processes where numerous WW domains are involved. Understanding these sequence/affinity relationships may help to sort out some of the confusing cell biological data on these interactions. The importance of these tuned affinities is shown by the experiments on YAP/LATS phosphorylation in Figure 5.

Essential revisions:

The reviewers feel that the manuscript is rather dense and hard to digest, and would benefit from reorganizing it around the key take-home message that affinity enhancements provided by tandem WW and PY motifs enable specificity for specific proteins, given the relatively weak binding of single WW domains to PY peptides. This includes the effect of the spacing, rigidity and sequence of the WW-domain linker, and also that the tandem also enables binding to non-canonical PY motifs and flanking sequences. It may help to put some of the detail into the figure legends.

1) Although the findings are exciting and important, the idea that is illustrated so clearly here is not new. The finding that tandem (bivalent) interactions are higher in affinity than monovalent interactions has been discussed for decades. It is commonly accepted that individual WW domains can bind promiscuously, albeit weakly, and that many proteins use numerous or tandem WW domains to enhance affinity and specificity (e.g. see overview in Dodson et al., 2015). A similar mechanism of affinity enhancement has been demonstrated for tandem SH2 domains, e.g., tandem SH2 domains found in proteins such as ZAP-70 can bind to their native TAMs with low nanomolar affinity, and the spacing of the motifs in the TAMs matter. The authors should include a discussion of such prior work, to provide context for their results. It would also be useful to discuss the expected affinity enhancements from a theoretical perspective. Huan-Xiang Zhou at Florida State has written a lot on this. Is a 1000X enhancement just from having two adjacent domains rigid in a binding competent form (KIBRA) compared to flexible (YAP) expected? Assuming the binding competent form is energetically accessible to YAP, the question is, what fraction of the total conformations does it represent? If it is 1/1000, then the observed result is expected. If it is 1/10 then it is not. Obviously, the number of conformations in YAP (with disordered linker) will depend on its linker length, but Zhou et al. have decent models for this.

2) In the text you describe large specificity enhancements for ordered tandem WW modules like KIBRA. It would be helpful to define what you mean by specificity, and give specific comparisons that demonstrate the specificity enhancement. It seems likely that the specificity being discussed is large differences in affinities of the tandem for different PY2 motifs compared to small differences in affinities of the corresponding single WW motifs for respective PY1 sequences. If so, please clarify and expand, and if not, please describe what is meant by specificity and what observations support the claim

3) It is not made clear whether the interactions of the tandem WW domains seen in the ligand complexes are present in the absence of ligand. Apart from the inability to crystallize the YAP complexes without fusions (a fairly weak argument), the authors should discuss whether there is evidence for flexibility between them when not bound. Have NMR or SAXS analyses been done, for example? Also, given that ITC was used to measure the affinities, it would be useful to know if any insights into can be gleaned from the enthalpy/entropy breakdowns. For example, if the formation of the complex immobilizes the two domains, one might expect to see a relatively unfavorable entropy of binding (in addition to possible unfavorable entropy of binding due to immobilizing flexible peptide ligands).

[Editors' note: further revisions were requested prior to acceptance, as described below.]

Thank you for resubmitting your work entitled "Decoding WW Domain Tandem-mediated Target Recognitions in Tissue Growth and Cell Polarity" for further consideration at *eLife*. Your revised article has been favorably evaluated by John Kuriyan (Senior Editor) and a Reviewing Editor.

The manuscript has been improved but there are some remaining issues that need to be addressed before acceptance, as outlined below:

1) For the response to the essential revision 1, the authors have added a general statement in the last paragraph of subsection “Origin of the exquisite target binding specificity of WW tandems” and referenced a review. It would be appropriate to add the simple example provided in the response to the Discussion section; even if the paper is aimed largely at cell biologists, the broader implications of the work for readers studying protein-protein interactions are of interest, and a short paragraph summarizing the thermodynamic principles seems like a reasonable addition.

2) Subsection “YAP WW tandem adopts a very different structure and target binding mode compared to the KIBRA WW tandem”: Need to specify that Yorkie and Yap are orthologs.

3) In the Figure 2B legend, please add something like "…KIBRA/Dendrin complex described earlier (Ji et al., 2019), so the WW supramolecular complex is formed from two different polypeptides in this structure. However, in solution KIBRA and AMOT form a 1:1 complex."

---

## [Author Response]

Essential revisions:The reviewers feel that the manuscript is rather dense and hard to digest, and would benefit from reorganizing it around the key take-home message that affinity enhancements provided by tandem WW and PY motifs enable specificity for specific proteins, given the relatively weak binding of single WW domains to PY peptides. This includes the effect of the spacing, rigidity and sequence of the WW-domain linker, and also that the tandem also enables binding to non-canonical PY motifs and flanking sequences. It may help to put some of the detail into the figure legends.

Thanks for the wonderful summary of the take-home message of our study and great suggestion on the presentation of our results. During the revision, we have made additional efforts to reorganize and simplify the manuscript in order to better present the key message. We have also revised figures and figure legends by including more details so that each figure together with its legend can clearly convey a core message.

1) Although the findings are exciting and important, the idea that is illustrated so clearly here is not new. The finding that tandem (bivalent) interactions are higher in affinity than monovalent interactions has been discussed for decades. It is commonly accepted that individual WW domains can bind promiscuously, albeit weakly, and that many proteins use numerous or tandem WW domains to enhance affinity and specificity (e.g. see overview in Dodson et al., 2015). A similar mechanism of affinity enhancement has been demonstrated for tandem SH2 domains, e.g., tandem SH2 domains found in proteins such as ZAP-70 can bind to their native TAMs with low nanomolar affinity, and the spacing of the motifs in the TAMs matter. The authors should include a discussion of such prior work, to provide context for their results. It would also be useful to discuss the expected affinity enhancements from a theoretical perspective. Huan-Xiang Zhou at Florida State has written a lot on this. Is a 1000X enhancement just from having two adjacent domains rigid in a binding competent form (KIBRA) compared to flexible (YAP) expected? Assuming the binding competent form is energetically accessible to YAP, the question is, what fraction of the total conformations does it represent? If it is 1/1000, then the observed result is expected. If it is 1/10 then it is not. Obviously, the number of conformations in YAP (with disordered linker) will depend on its linker length, but Zhou et al., have decent models for this.

We share the same view as that raised by the reviewers. The concept of “tandem (bivalent) interactions are higher in affinity than monovalent interactions” is indeed well established. This concept is rooted in the Second Law of Thermodynamics. For a tandem/bivalent system, the binding energy can be written as ΔG_total_ = ΔG_1_ + ΔG_2_ - DG_c_; where ΔG_1_ and ΔG_2_ represent the binding free energy of the two individual sites and ΔG_c_ can be viewed as amount of energy penalty (or conformational entropy loss; also see Zhou and Gilson, 2009, review) due to non-perfect conformational coupling of the two binding sites. For a system with perfect conformational coupling, ΔG_c_ would be approachingzero. For a system without any conformational coupling, the two bind sites can be viewed as completely separate (i.e. the stronger binding site prevails). ΔG_c_ often falls in between the above two extreme conditions in real biological systems. For example, in the YAP/Dendrin system, the binding affinity for the YAP WW tandem to PY23 of Dendrin is ~400 nM in *K_d_*. A perfect coupling would give a *K_d_* of ~0.6 nM (16 μM x 37 μM; the binding affinities of each WW domain to PY23 are presented in Figure 1—figure supplement 1B). For the KIBRA WW tandem, the situation is even more complicated as its WW2 is only partially folded in the absence of ligands (Ji et al., 2019). The total free energy will also include a binding-induced domain folding energy.

Since the general thermodynamic concept on the bivalent system is so well-established, we did not elaborate this in any detail in our manuscript. Additionally, the manuscript at the current stage has already contained a huge amount of information, we decided not to make any attempt in dissecting the thermodynamic terms of the binding events. Additionally, we intend to make this paper also accessible to colleagues outside the field of biochemistry and biophysics, so we have been trying to minimize thermodynamic details of the bindings investigated. We have cited the nice review paper by Zhou and Gilson (2009) in the revised manuscript (at the end of the subsection “Origin of the exquisite target binding specificity of WW tandems”) by stating “The above findings are consistent with well-established thermodynamic principles governing multivalent protein/ligand interactions (see Zhou and Gilson (2009) for a review)”.

What is not known for the WW tandem-mediated target binding system is how such thermodynamic principle is used. Our study uncovered that certain WW tandems have strong couplings, certain ones have weak coupling, and some are essentially with very little coupling. We have also provided structural bases for such different modes of couplings. The findings presented in our study should be valuable for colleagues working on WW tandem-related proteins. Having said this, we totally agree that our study does *not* uncover new thermodynamic concept for the bivalent biomolecular interactions. Accordingly, we did not make any claim on the discovery of thermodynamics concepts of the bindings in our manuscript. Instead, we have used the well-established thermodynamic principle to explain and rationalize the findings for the WW tandem/target interactions investigated in our study.

2) In the text you describe large specificity enhancements for ordered tandem WW modules like KIBRA. It would be helpful to define what you mean by specificity, and give specific comparisons that demonstrate the specificity enhancement. It seems likely that the specificity being discussed is large differences in affinities of the tandem for different PY2 motifs compared to small differences in affinities of the corresponding single WW motifs for respective PY1 sequences. If so, please clarify and expand, and if not, please describe what is meant by specificity and what observations support the claim

This is a great point! Binding specificity is related but does not equal to binding affinity. For a mixture of many proteins capable of forming different complexes in a test tube, the binding specificity of a particular pair is determined by their binding affinity relative to other interactions and the concentration of each component (as we mentioned in the second paragraph in the Discussion section). The binding specificity in living cells is even more complicated due to additional factors such as spatial segregations of different components. This point is very important to the WW domain proteins covered in the current study, as the vast majority of WW domain-mediated target interaction studies published in the literature used either overexpression or removal (knockdown or knockout) approaches. Very often a detected interaction after such protein manipulations may not reflect the true cellular interactions under their native conditions.

For the WW tandems studied here, their target binding affinity is highly amino acid sequence specific. A high affinity binding requires two PY motifs to be separated by two and only two residues. Residues between the two PY motifs and sequences following the second PY motif further enhance binding affinity to WW tandems. Thus, a super strong WW tandem binding target tends to have very unique amino acid sequence features (i.e. such sequences do not frequently appear in many proteins), and we view this as the intrinsic specificity for WW tandem/target interactions dictated by the structure of each WW tandem and the unique sequences of its PY motif-containing targets.

But for dissecting the ligand preferences for the PY2 motif in Figure 6, the measured bindings for different residues in the motif simply reflect binding affinities instead of specificities. We have revised the description following the reviewer’s comment.

3) It is not made clear whether the interactions of the tandem WW domains seen in the ligand complexes are present in the absence of ligand. Apart from the inability to crystallize the YAP complexes without fusions (a fairly weak argument), the authors should discuss whether there is evidence for flexibility between them when not bound. Have NMR or SAXS analyses been done, for example? Also, given that ITC was used to measure the affinities, it would be useful to know if any insights into can be gleaned from the enthalpy/entropy breakdowns. For example, if the formation of the complex immobilizes the two domains, one might expect to see a relatively unfavorable entropy of binding (in addition to possible unfavorable entropy of binding due to immobilizing flexible peptide ligands).

The conformation of the KIBRA WW tandem in the absence of ligand has been characterized in detail in our previous NMR-based study (Ji et al., 2019). WW1 of the KIBRA WW tandem is well folded and WW2 of the tandem is only partially folded. The conformation of the apo-YAP WW tandem has also been characterized by NMR spectroscopy (Webb et al., 2011). It was shown that there is minimal coupling between the two WW domains based on NMR chemical shift-based analysis. We have added the above information in the revised manuscript and cited the relevant references.

We have also looked into the thermodynamic parameters of the YAP WW tandem/Dendrin PY23 interactions (see Author response image 1). The data show that formation of the complex WW12/PY23 is indeed somewhat entropically unfavorable compared to the individual WW domain bindings (ΔS for the tandem is -93.3 cal/mol·K, which is larger than the sum of the ΔS for the two isolated WW domain bindings, which is -127.9 cal/mol·K).

**Author response image 1. respfig1:** ITC experiments of YAP WW12 tandem and individual WW1 or WW2 titrated to Dendrin PY23. The fitted *K_d_* and the corresponding calculated entropy change are indicated in the figure for each titration.

[Editors' note: further revisions were requested prior to acceptance, as described below.]

The manuscript has been improved but there are some remaining issues that need to be addressed before acceptance, as outlined below:1) For the response to the essential revision 1, the authors have added a general statement in the last paragraph of subsection “Origin of the exquisite target binding specificity of WW tandems” and referenced a review. It would be appropriate to add the simple example provided in the response to the Discussion section; even if the paper is aimed largely at cell biologists, the broader implications of the work for readers studying protein-protein interactions are of interest, and a short paragraph summarizing the thermodynamic principles seems like a reasonable addition.

We have included the following text in the Discussion section: “The enhanced PY-motif containing target binding afforded by WW domains connected in tandem is in line with the Second Law of Thermodynamics. […] The total free energy will also include a binding-induced domain folding energy.”

2) Subsection “YAP WW tandem adopts a very different structure and target binding mode compared to the KIBRA WW tandem”: Need to specify that Yorkie and Yap are orthologs.

We have specified that Yorkie is the Yap ortholog in *Drosophila*.

3) In the Figure 2B legend, please add something like "…KIBRA/Dendrin complex described earlier (Ji et al., 2019), so the WW supramolecular complex is formed from two different polypeptides in this structure. However, in solution KIBRA and AMOT form a 1:1 complex."

We have added the description in the legend of Figure 2B as suggested. The added text reads as “Figure 2B: In the crystals of the KIBRA/AMOT complex, WW12 adopts a domain-swapped dimer similar to the crystals of the KIBRA/Dendrin complex described earlier (Ji et al., 2019), so the KIBRA/AMOT supramolecular complex is formed from two different polypeptides in this structure. However, in solution KIBRA and AMOT form a 1:1 complex.”